# A G358S mutation in the *Plasmodium falciparum* Na⁺ pump PfATP4 confers clinically-relevant resistance to cipargamin

Deyun Qiu [1,10], Jinxin V. Pei [1,10], James E. O. Rosling [1,10], Vandana Thathy[2], Dongdi Li[1], Yi Xue[1], John D. Tanner [1], Jocelyn Sietsma Penington [3], Yi Tong Vincent Aw[1], Jessica Yi Han Aw[1], Guoyue Xu [4], Abhai K. Tripathi [4], Nina F. Gnadig[2], Tomas Yeo [2], Kate J. Fairhurst[2], Barbara H. Stokes [2], James M. Murithi[2], Krittikorn Kümpornsin[5], Heath Hasemer[1], Adelaide S. M. Dennis[1], Melanie C. Ridgway [1], Esther K. Schmitt[6], Judith Straimer[7], Anthony T. Papenfuss [3,8], Marcus C. S. Lee [5], Ben Corry [1], Photini Sinnis [4], David A. Fidock [2,9], Giel G. van Dooren [1], Kiaran Kirk[1] & Adele M. Lehane [1] ✉

Diverse compounds target the *Plasmodium falciparum* Na⁺ pump PfATP4, with cipargamin and (+)-SJ733 the most clinically-advanced. In a recent clinical trial for cipargamin, recrudescent parasites emerged, with most having a G358S mutation in PfATP4. Here, we show that PfATP4^G358S parasites can withstand micromolar concentrations of cipargamin and (+)-SJ733, while remaining susceptible to antimalarials that do not target PfATP4. The G358S mutation in PfATP4, and the equivalent mutation in *Toxoplasma gondii* ATP4, decrease the sensitivity of ATP4 to inhibition by cipargamin and (+)-SJ733, thereby protecting parasites from disruption of Na⁺ regulation. The G358S mutation reduces the affinity of PfATP4 for Na⁺ and is associated with an increase in the parasite's resting cytosolic [Na⁺]. However, no defect in parasite growth or transmissibility is observed. Our findings suggest that PfATP4 inhibitors in clinical development should be tested against PfATP4^G358S parasites, and that their combination with unrelated antimalarials may mitigate against resistance development.

Malaria, an ancient mosquito-borne disease caused by *Plasmodium* parasites, killed an estimated 627,000 people in 2020, of whom approximately 500,000 were children under the age of 5 years[1]. Among the *Plasmodium* species that cause disease in humans, *Plasmodium falciparum* causes the vast majority of deaths[1]. *P. falciparum* parasites with reduced susceptibility to several of the latest first-line malaria treatments, which consist of an artemisinin derivative paired with a quinoline-related compound, are spreading[2,3]. The situation is precarious, as the widespread failure of artemisinin-based combination therapies would render malaria more difficult to treat[4,5]. Thus, to

safeguard malaria treatment, new antimalarial chemotypes with novel modes of action are required, along with strategies to protect them from resistance.

In the last 15 years, millions of compounds have been screened for their ability to kill asexual blood-stage *P. falciparum* parasites[6–8]. This has led to the discovery of promising new antimalarial drug classes, including the spiroindolones[9]. In an attempt to identify their target, in vitro evolution experiments were performed in which *P. falciparum* parasites were exposed to spiroindolones until parasites emerged that displayed low-level resistance. Resistance was

associated with mutations in the gene encoding the P-type ATPase PfATP4, which was localised to the parasite plasma membrane[9]. PfATP4 was originally thought to be a $Ca^{2+}$ transporter[10]; however, Spillman et al.[11] provided evidence that it is in fact a $Na^+$ transporter that exports $Na^+$ from the parasite cytosol, whilst importing $H^+$ equivalents. Spiroindolones cause a rapid increase in the $[Na^+]$ in the parasite cytosol ($[Na^+]_{cyt}$)[11]. Parasites with mutations in PfATP4, showing low-level resistance to spiroindolones, were found to be less sensitive to the $Na^+$-dysregulating effects of spiroindolones than their parents, and to have a higher resting $[Na^+]_{cyt}$[11]. These observations are consistent with the resistance-conferring mutations impacting the $Na^+$-efflux function of PfATP4[11].

As well as causing a rise in $[Na^+]_{cyt}$ (and thereby dissipating the inward $[Na^+]$ gradient across the parasite plasma membrane), spiroindolones cause a variety of other physiological perturbations including: an alkalinisation of the parasite cytosol, which *increases* the pH gradient across the parasite plasma membrane[11]; an increase in the volume of parasites and parasitised erythrocytes, attributable to the osmotic consequences of the $[Na^+]_{cyt}$ increase[12]; a reduction in cholesterol extrusion from the parasite plasma membrane resulting from the increase in $[Na^+]_{cyt}$[13]; and an increase in the rigidity of erythrocytes infected with ring-stage parasites[14]. Spiroindolones have also been shown to inhibit a $Na^+$-dependent, pH-sensitive ATPase in parasite membrane preparations, which likely corresponds to PfATP4[11,15]. Together, the available data are consistent with PfATP4 functioning as an ATP-dependent transporter on the parasite plasma membrane, extruding $Na^+$ from the parasite while importing $H^+$.

In contrast to *P. falciparum* parasites, which experience a high external $Na^+$ concentration for the majority of their 48 h asexual life cycle[16] and for which a reduction of *pfatp4* expression is deleterious to growth[17], *T. gondii* parasites are only known to be exposed to a high external $Na^+$ concentration for the brief period in their lytic cycle in which they are extracellular, and can survive and proliferate when their ATP4 homologue (TgATP4) is not expressed[18]. Physiological studies with *T. gondii* parasites lacking TgATP4 expression, or treated with ATP4 inhibitors such as cipargamin, have provided further evidence that ATP4 proteins export $Na^+$ while importing $H^+$[18].

In the years since the discovery of the spiroindolones, a large number of chemically diverse compounds have been found to perturb parasite physiology in the same manner as the spiroindolones[19–25]. These include the dihydroisoquinolone (+)-SJ733[23], which has recently been tested in humans[26], the pyrazoleamide PA21A050[25], and multiple less clinically advanced compounds[19–22,24]. The spiroindolone cipargamin (previously referred to as KAE609 or NITD609) is the most clinically advanced of the compounds proposed to target PfATP4.

Oral formulations of cipargamin have been tested in multiple Phase 1 and Phase 2 clinical trials, with testing underway with an intravenous formulation suitable for the treatment of severe malaria (reviewed in ref. 27). Among the important attributes of cipargamin are (i) its ability to achieve rapid clearance of both *P. falciparum* and *P. vivax* parasites with minimal side effects[28]; (ii) its favourable pharmacokinetic properties, which may make it suitable for a simplified dosing schedule[27]; and (iii) its activity against sexual stage parasites in laboratory studies, which may translate into transmission-blocking activity in the field[29–31].

Another important consideration for compounds in clinical development is the risk of selection for parasite resistance[32]. There is significant variability in the frequency with which *P. falciparum* parasites acquire resistance to different compounds. For example, atovaquone resistance occurs at a high frequency of $\sim 10^{-5}$ under certain conditions in some strains (i.e. 1 in $10^5$ parasites acquire resistance)[33], whereas for other compounds it has not yet been possible to generate resistance in the laboratory (e.g. INE963[34]). It has been known since 2010 that long-term drug-exposure experiments with cipargamin can yield PfATP4-mutant parasites with low-level resistance[9]. The

frequency with which parasites acquired low-level resistance when exposed continuously to 2.5 nM cipargamin varied from $\sim 2 \times 10^{-8}$ to $1 \times 10^{-7}$, depending on the genetic background of the strain[35]. In vitro evolution experiments have also been performed with other spiroindolones and with numerous chemically distinct compounds that also display the physiological hallmarks of PfATP4 inhibition, yielding resistant parasites with mutations in *pfatp4* in each case[20,21,23–25]. More than 40 different resistance-associated SNPs have been reported in *pfatp4*, with most individual parasite lines having a single mutation in PfATP4, a small number having two mutations, and one reported to have three mutations[9,20,21,23–25,35]. Cross-resistance to structurally unrelated PfATP4 inhibitors has been demonstrated for multiple distinct PfATP4-mutant parasites[20,21,23,24,36].

It has been suggested that the frequency of resistance to PfATP4 inhibitors may be lower in the field than it is in vitro, as a result of the rapid speed by which the inhibitors kill parasites in vivo and fitness costs associated with resistance[23]. In vitro growth competition experiments with asexual parasites have been reported for two (+)-SJ733-resistant PfATP4-mutant *P. falciparum* lines (PfATP4[L350H] and PfATP4[P996T]), and both of these mutants were found to have a growth disadvantage relative to their parents[23]. However, in a different study, PfATP4-mutant parasites selected for resistance to the aminopyrazole GNF-Pf4492 (PfATP4[A211T], PfATP4[I203L/P990R] and PfATP4[A187V]) were reported to proliferate at the same rate as their parents[20]. Thus, it appears that some resistance-conferring PfATP4 mutations confer a growth disadvantage while others do not.

To date, the level of resistance that has been achieved in vitro for cipargamin has been modest. At least 18 parasite lines with different *pfatp4* mutations have been reported after in vitro evolution experiments with cipargamin[9,35]. The 50% growth inhibitory concentrations ($IC_{50}$s) for cipargamin ranged from 1.5 to 24.3 nM for the mutant parasites, compared to 0.4 to 1.1 nM for the parental lines used in these experiments. If parasites with a similar level of resistance to the in vitro selected parasites were to emerge in the field, it is likely that cipargamin would remain effective against them. In clinical trials, cipargamin has been found to reach supramicromolar concentrations in human plasma for sustained periods[28,37–39].

However, in a recent Phase 2a clinical trial for oral cipargamin in patients with uncomplicated malaria, recrudescent parasites emerged[40]. In most (22/25) instances of recrudescence, a G358S mutation in PfATP4 was observed[40]. All patients with recrudescent parasites were successfully treated with artemether-lumefantrine[40]. In this study, we investigated parasites with the G358S mutation in PfATP4, generated through in vitro selections with cipargamin or genetic engineering. We investigated the effect of the G358S mutation on the function and chemical sensitivity of PfATP4, its effect on the parasite's physiology, growth, and susceptibility to a variety of PfATP4 inhibitors and unrelated antimalarial drugs, and its impact on the ability of the parasite to complete its life cycle.

## Results

### *P. falciparum* parasites can acquire high-level resistance to cipargamin in vitro

To determine whether *P. falciparum* parasites with high-level resistance to cipargamin can be generated through in vitro evolution, we exposed two independent parasite cultures to incrementally increasing concentrations of cipargamin over the course of four months (Fig. 1a). We commenced this experiment with a parasite line that had previously been selected for cipargamin resistance (NITD609-R[Dd2] clone#2[9]; referred to here as Dd2-PfATP4[T418N,P990R]). This line displays low-level resistance to cipargamin (Fig. 1b, Fig. 2, Supplementary Table 1) and has two PfATP4 mutations (T418N and P990R) that are not present in its parental Dd2 strain.

At the end of the drug-exposure period, parasites from both cultures were highly resistant to cipargamin, with $IC_{50}$ values of

**a**

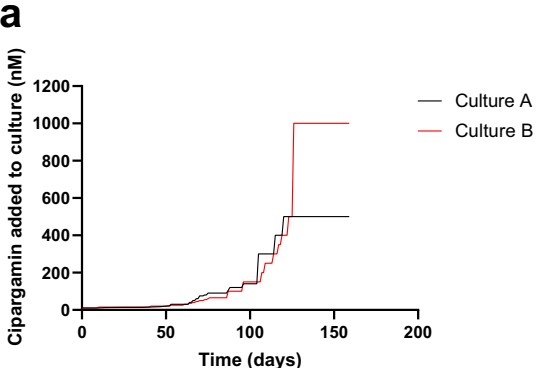

**b**

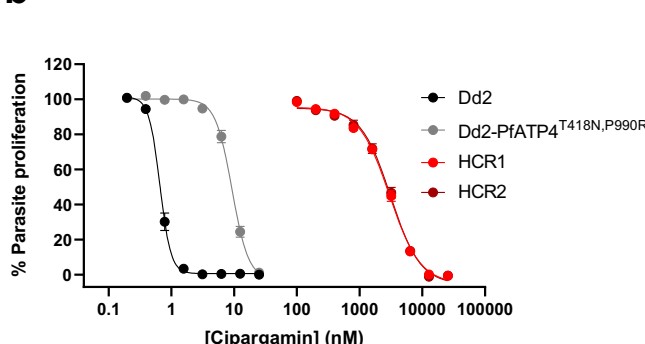

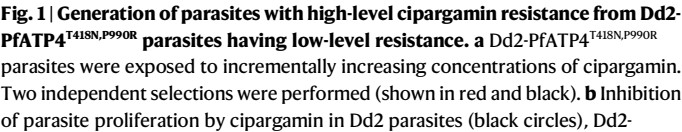

**Fig. 1 | Generation of parasites with high-level cipargamin resistance from Dd2-PfATP4$^{T418N,P990R}$ parasites having low-level resistance. a** Dd2-PfATP4$^{T418N,P990R}$ parasites were exposed to incrementally increasing concentrations of cipargamin. Two independent selections were performed (shown in red and black). **b** Inhibition of parasite proliferation by cipargamin in Dd2 parasites (black circles), Dd2-PfATP4$^{T418N,P990R}$ parasites (grey circles), and highly cipargamin-resistant (HCR) parasites (clone #1 in red circles and clone #2 in dark red circles). The data shown are the mean ($\pm$ SEM) from 16 independent experiments, each performed on different days. All lines were tested in parallel. Where not shown, error bars fall within the symbols. Source data are provided as a Source Data file.

$5.9 \pm 0.2$ µM (culture A; mean $\pm$ SEM, $n = 3$) and $6.2 \pm 0.2$ µM (culture B; mean $\pm$ SEM, $n = 3$). By comparison, the initial Dd2-PfATP4$^{T418N,P990R}$ parasite line had an IC$_{50}$ value for cipargamin below 10 nM and its Dd2 parent had a subnanomolar IC$_{50}$ value (Supplementary Table 1). We subjected the cipargamin-selected cultures to limiting dilution and selected clonal lines ('highly cipargamin-resistant (HCR)' clones 1 and 2, from culture B) for further characterisation. The HCR1 and HCR2 parasites were highly resistant to cipargamin, having IC$_{50}$ values 315–326 fold higher than that of their Dd2-PfATP4$^{T418N,P990R}$ parents, and 4200–4280 fold higher than that of wild-type Dd2 parasites (mean, $n = 16$; Fig. 2; Supplementary Table 1).

We also attempted to generate highly cipargamin-resistant parasites using a single-step protocol in which parasites were exposed to a single high concentration of cipargamin. We exposed two 150 mL cultures, each containing ~$6.6 \times 10^9$ Dd2-PfATP4$^{T418N,P990R}$ parasites, to a ~$10 \times$ IC$_{50}$ concentration of cipargamin (102 nM). We did not recover viable resistant parasites using this approach with Dd2-PfATP4$^{T418N,P990R}$ parasites, suggesting that the acquisition of high-level resistance to cipargamin occurs at a low frequency.

We then attempted to generate highly cipargamin-resistant parasites using a single-step procedure with a *P. falciparum* Dd2 line with a hypermutator phenotype. These parasites, referred to here as 'Dd2-Polδ', were modified using CRISPR-Cas9 to introduce two mutations into the gene encoding the DNA polymerase delta subunit in order to impair the protein's proofreading function[41]. Dd2-Polδ parasites (three independent cultures, each containing $1.5 \times 10^8$ parasites) were exposed to cipargamin at a single concentration of 25 nM, with one additional culture containing $1.5 \times 10^8$ parasites exposed to 100 nM cipargamin. Viable parasites were observed in one of the cultures containing 25 nM cipargamin ('cipargamin-selected Culture 1') after 17 days. Resistant parasites did not emerge in the other three cultures within 25 days and these cultures were discarded. A clone from cipargamin-selected Culture 1 was then obtained and characterised further. This clone was found to be highly resistant to cipargamin (Fig. 2), with an IC$_{50}$ value $1082 \pm 138$ fold (mean $\pm$ SEM, $n = 7$) higher than that of its Dd2-Polδ parent (Supplementary Table 1).

We also used Dd2-Polδ parasites to select for parasites resistant to additional PfATP4-associated chemotypes, in each case performing single-step selections with a concentration equating to $6$–$15 \times$ the IC$_{50}$ for the compound. First, we exposed Dd2-Polδ parasites (two independent cultures, each containing $2 \times 10^8$ parasites) to 250 nM (+)-SJ733. After 16 days, viable parasites were observed in one of the cultures ('(+)-SJ733-selected Culture 1'; the other culture had no viable

parasites on Day 25 and was discarded). We also attempted to generate high-level resistance to two compounds from the Medicines for Malaria Venture's (MMV's) 'Malaria Box', namely MMV665949 and MMV006656. These compounds are structurally unrelated to the spiroindolones, dihydroisoquinolones, pyrazoleamides, or each other, but display the hallmarks of PfATP4 inhibition: an increase in [Na$^+$]$_{cyt}$ and pH$_{cyt}$[24] and inhibition of Na$^+$-ATPase activity in parasite membrane preparations[15]. We exposed Dd2-Polδ parasites to a single high concentration of 50 µM MMV665949 or 5 µM MMV006656 (two independent cultures, each containing $2 \times 10^8$ parasites for each compound). For MMV665949, resistant parasites emerged from one of the cultures ('MMV665949-selected Culture 1') on Day 17. No parasites were observed in the second MMV665949-containing culture within 25 days and the culture was discarded. For MMV006656, no viable parasites were observed and the cultures were discarded on Day 42.

## High-level cipargamin resistance is conferred by a G358S mutation in PfATP4

We extracted genomic DNA from the Dd2-PfATP4$^{T418N,P990R}$ cultures selected with cipargamin (cultures A and B) and from four clonal lines (HCR1-4) obtained from culture B. Sequencing the *pfatp4* gene revealed the same set of mutations in each case (relative to the Dd2 sequence): those encoding the T418N and P990R mutations present in the Dd2-PfATP4$^{T418N,P990R}$ parental line and a third mutation (G1072A) that codes for a G358S mutation in the PfATP4 protein.

We subjected Dd2, Dd2-PfATP4$^{T418N,P990R}$, HCR1 and HCR2 to whole-genome sequencing. This revealed a duplication in chromosome 12 in HCR1 and HCR2 that was not present in Dd2 or Dd2-PfATP4$^{T418N,P990R}$, in a region (spanning the chromosomal positions 520–556 kb) covering nine genes including *pfatp4*. The *pfatp4* mutation coding for G358S was detected at a frequency of ~50% in both HCR1 and HCR2 (Supplementary Fig. 1), suggesting that the duplication of *pfatp4* occurred before the mutation, and that the mutation was present in only one of the two copies of the gene. Dd2-PfATP4$^{T418N,P990R}$, HCR1 and HCR2 were all confirmed to have the mutations coding for T418N and P990R in PfATP4. Thus, the HCR1 and HCR2 parasites have two copies of *pfatp4*, one encoding the PfATP4$^{T418N,P990R}$ variant present in the parental line and the other encoding the triple-mutant variant PfATP4$^{G358S,T418N,P990R}$. Four other SNPs in different genes, which we considered unlikely to contribute to cipargamin resistance, were also noted in HCR1 and/or HCR2 (Supplementary Note). A reduction in the degree of amplification of a region on chromosome 5 containing *pfmdr1* was also observed in HCR2 (Supplementary Note, Supplementary Fig. 1).

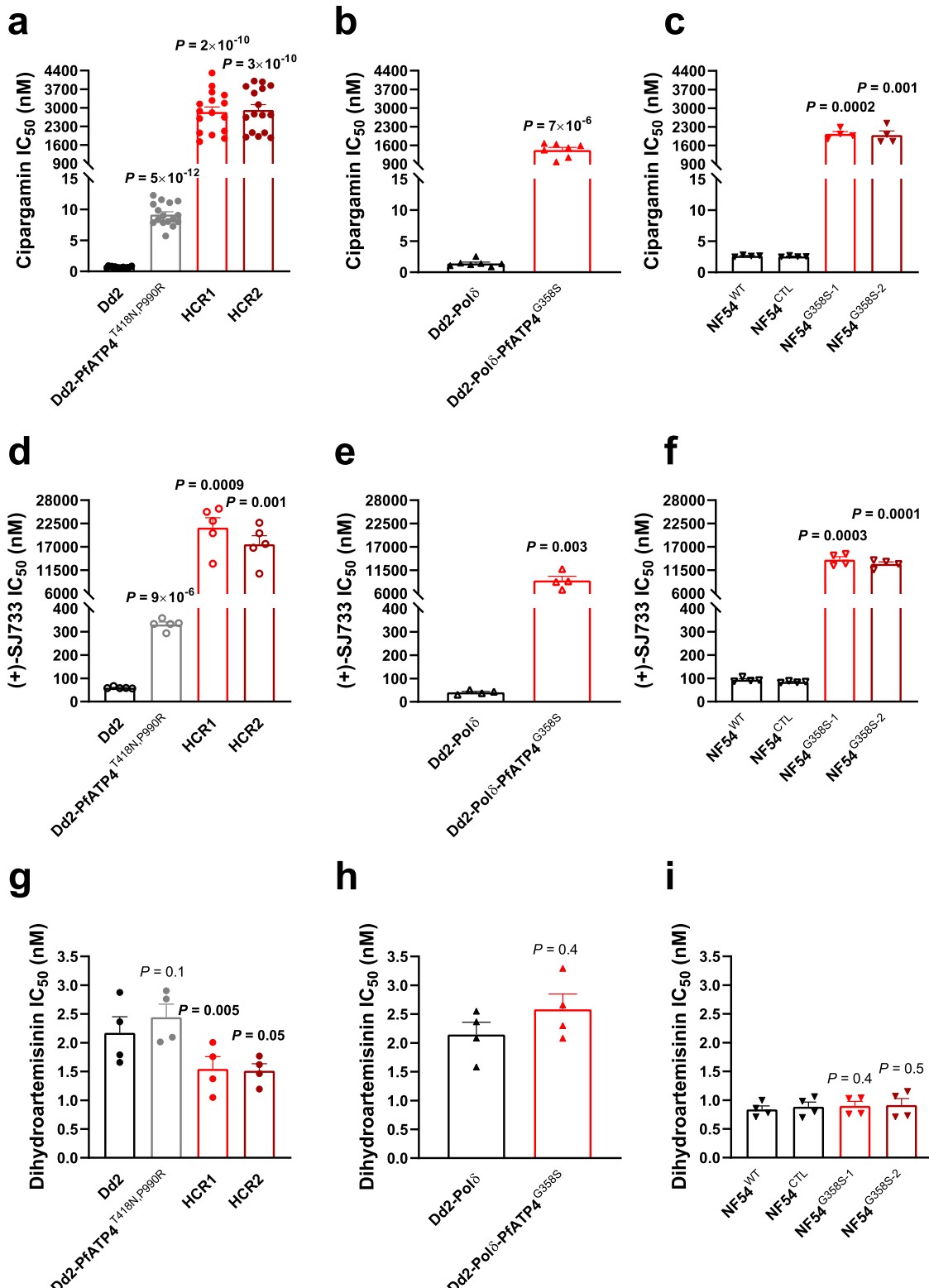

**Fig. 2 | Parasites with the G358S mutation in PfATP4 display a high level of resistance to cipargamin and (+)-SJ733.** IC$_{50}$ values are shown for cipargamin (**a**–**c**), (+)-SJ733 (**d**–**f**) and dihydroartemisinin (**g**–**i**) against HCR parasites (clone #1 in red and clone #2 in dark red) generated through in vitro evolution with the Dd2-PfATP4$^{T418N,P990R}$ line (grey; with data for its Dd2 parent also shown in black; **a**, **d**, **g**, circles), Dd2-Polδ-PfATP4$^{G358S}$ (red) parasites generated through in vitro evolution with the Dd2-Polδ line (black; **b**, **e**, **h**, triangles), and two NF54$^{G358S}$ lines (1 in red and 2 in dark red) generated using CRISPR-Cas9 with the NF54$^{WT}$ line (black; along with the matched recombinant control line NF54$^{CTL}$ in black; (**c**, **f**, **i**), upside

down triangles). Within each panel, all lines were tested in parallel in each experiment. The symbols show the data from individual experiments and the bars show the mean + SEM. The data are from the following number of independent experiments, performed on different days: 16 (**a**), 7 (**b**), 4 (**c**), 5 (**d**), 4 (**e**), 4 (**f**), 4 (**g**), 4 (**h**) and 4 (**i**). For each compound, the IC$_{50}$ values for each cipargamin-resistant line were compared with those of its direct parent using two-tailed paired $t$ tests ($P$ values ≤ 0.05 indicate statistical significance and are shown in bold). Source data are provided as a Source Data file.

We also extracted genomic DNA from the Dd2-Polδ bulk cultures selected with cipargamin (cipargamin-selected Culture 1) or (+)-SJ733 ((+)-SJ733-selected Culture 1), as well as from a clone derived from cipargamin-selected Culture 1, and sequenced the *pfatp4* gene. In all cases we observed a single mutation in the *pfatp4* gene that was not present in the Dd2-Polδ parents—a G1072A nucleotide mutation that codes for the G358S mutation in PfATP4.

We used CRISPR-Cas9 in the transmission-competent NF54 strain ('NF54$^{WT}$') to generate *P. falciparum* parasites that harbour the G358S mutation in PfATP4 (Supplementary Fig. 2). The gene-edited NF54 control line (NF54$^{CTL}$) harbours 17 silent binding-site mutations at the guide RNA cleavage site within the *pfatp4* locus. Three *pfatp4* mutant lines, NF54$^{G358S-1}$ and NF54$^{G358S-3}$ that were technical replicates from one transfection, and NF54$^{G358S-2}$ that was generated from an independent transfection, have a mutation coding for the G358S change in the PfATP4 protein, along with the 17 silent binding-site mutations. Whole-genome sequencing confirmed the presence of a mutation coding for the G358S change in PfATP4 in the NF54$^{G358S-1}$ and NF54$^{G358S-2}$ lines that was not present in the NF54$^{WT}$ or NF54$^{CTL}$ lines (Supplementary Tables 2–4). The NF54$^{G358S-1}$ and NF54$^{G358S-2}$ parasites were highly resistant to cipargamin (Fig. 2), with IC$_{50}$ values > 750-fold greater than those for NF54$^{WT}$ and NF54$^{CTL}$ (Supplementary Table 5). NF54$^{G358S-3}$ parasites were also highly resistant to cipargamin, with an IC$_{50}$ value of $2.2 \pm 0.7$ μM (mean $\pm$ SEM, $n = 3$).

## Response of parasites harbouring the PfATP4 G358S mutation to a variety of antiplasmodial agents

In addition to being highly resistant to cipargamin, the HCR1 and HCR2 parasites (both of which harbour two PfATP4 variants: PfATP4$^{T418N,P990R}$ and PfATP4$^{G358S,T418N,P990R}$) were highly resistant to the dihydroisoquinolone (+)-SJ733 (Fig. 2) and the pyrazoleamide PA21A050, with the IC$_{50}$ values for the HCR parasites being 54–66 fold and 22–23 fold higher, respectively, than those of their Dd2-PfATP4$^{T418N,P990R}$ parents, and 300–366 fold and 50–51 fold higher, respectively, than those of Dd2 (Supplementary Table 1). HCR1 and HCR2 parasites were slightly but significantly *more sensitive* to dihydroartemisinin (the compound formed in vivo from all clinically used artemisinin derivatives) than parental and Dd2 parasites (Fig. 2, Supplementary Table 1). HCR2 parasites were also slightly more resistant to chloroquine than the other parasites (Supplementary Table 1).

Our studies also revealed that the drug-selected Dd2-Polδ parasite clone ('Dd2-Polδ-PfATP4$^{G358S}$') was highly resistant to (+)-SJ733 (Fig. 2), with an IC$_{50}$ value $237 \pm 51$ fold (mean $\pm$ SEM, $n = 4$) higher than that of the Dd2-Polδ parent (Supplementary Table 1). Dd2-Polδ-PfATP4$^{G358S}$ parasites were moderately resistant to PA21A050 and MMV006656, with IC$_{50}$ values $5.3 \pm 0.1$ fold (mean $\pm$ SEM, $n = 4$) and $5.7 \pm 1.3$-fold (mean $\pm$ SEM, $n = 4$) higher, respectively, than those of the Dd2-Polδ parent (Supplementary Table 1). There was no significant change in their susceptibility to MMV665949, chloroquine or dihydroartemisinin (Supplementary Table 1). Thus, the G358S mutation in PfATP4 was associated with a decrease in parasite susceptibility to four out of five compounds for which there is evidence for PfATP4 inhibition, and did not affect parasite susceptibility to the unrelated drugs chloroquine or dihydroartemisinin.

As well as displaying a high level of resistance to cipargamin (Fig. 2, Supplementary Table 5), the NF54$^{G358S-1}$ and NF54$^{G358S-2}$ parasites were highly resistant to (+)-SJ733 (Fig. 2), with IC$_{50}$ values > 138-fold greater than those for NF54$^{WT}$ and NF54$^{CTL}$ (Supplementary Table 5). We also profiled our gene-edited NF54 lines and their parent against a variety of other antimalarials with unrelated modes of action. These included dihydroartemisinin (Fig. 2), KAF156, pyronaridine, piperaquine, monodesethyl-amodiaquine (the active metabolite of amodiaquine), and lumefantrine. We also tested the new clinical candidate INE963[34], for which the mode of action is not yet known. The G358S

mutation in PfATP4 did not affect parasite susceptibility to any of the antimalarials that do not target PfATP4 (Supplementary Table 5). All edited NF54 lines, including NF54$^{CTL}$, displayed mildly elevated increases of 1.4-1.5-fold ($P \leq 0.01$) in their INE963 IC$_{50}$ values. NF54$^{CTL}$ and NF54$^{G358S-2}$ parasite lines exhibited 0.7-fold ($P = 0.0008$) and 0.8-fold ($P = 0.05$) decreases respectively, in the IC$_{50}$ values when assayed against KAF156, compared with the NF54$^{WT}$ parental strain. The equivalent IC$_{50}$ values between the PfATP4 G358S and control wild-type lines provide evidence that this PfATP4 mutation had no impact on parasite susceptibility to INE963 or KAF156.

The G358S mutation in PfATP4 has been observed before in parasites selected with (+)-SJ733[23]. These parasites (referred to here as Dd2-B2-PfATP4$^{G358S}$) were earlier reported to display only a 5-fold resistance to (+)-SJ733[23], which is much lower than the fold differences in (+)-SJ733 sensitivity observed for parasites harbouring the G358S mutation in PfATP4 in this study. We therefore performed our own parasite proliferation assays with Dd2-B2-PfATP4$^{G358S}$ and its parent, and found that Dd2-B2-PfATP4$^{G358S}$ is highly resistant to both cipargamin and (+)-SJ733, with IC$_{50}$ values (mean $\pm$ SEM) of $1640 \pm 80$ nM ($n = 5$) and $23000 \pm 3200$ nM ($n = 4$), respectively. These IC$_{50}$ values are $795 \pm 98$ fold and $186 \pm 18$ fold higher than those for the Dd2-B2 parent ($2.17 \pm 0.24$ nM ($n = 5$; $P = 4 \times 10^{-5}$, two-tailed paired $t$ test) for cipargamin and $122 \pm 7$ nM for (+)-SJ733 ($n = 4$; $P = 0.006$, two-tailed paired $t$ test)).

The MMV665949-selected parasites (MMV665949-selected Culture 1; not cloned) were tested for their sensitivity to growth inhibition by MMV665949 and cipargamin. In paired experiments, the IC$_{50}$ value for MMV665949 was $10.7 \pm 2.0$ fold higher for MMV665949-selected Culture 1 (IC$_{50} = 44 \pm 7$ μM) than for the parental Dd2-Polδ parasites (IC$_{50} = 4.3 \pm 0.7$ μM) (mean $\pm$ SEM, $n = 4$; $P = 0.008$, two-tailed paired $t$ test). For cipargamin, MMV665949-selected Culture 1 had an IC$_{50}$ value of $1.42 \pm 0.30$ nM, compared to $0.57 \pm 0.02$ nM in paired experiments with the parental Dd2-Polδ parasites (mean $\pm$ SEM, $n = 3$; $P = 0.1$, two-tailed paired $t$ test). Thus, while the single-step selections with cipargamin and (+)-SJ733 both resulted in parasites with the G358S mutation in PfATP4 that were highly resistant to both compounds, the single-step selection with the structurally unrelated PfATP4-associated compound MMV665949 did not. The MMV665949-selected parasites were not characterised further in this study.

## Mechanistic basis for high-level resistance to cipargamin and (+)-SJ733 conferred by the G358S mutation in PfATP4

PfATP4 is required for the maintenance of a low [Na$^+$]$_{cyt}$ in *P. falciparum* parasites, with inhibition of the protein resulting in an increase in [Na$^+$]$_{cyt}$[11]. We investigated whether the G358S mutation in PfATP4 rendered parasites resistant to having their [Na$^+$]$_{cyt}$ dysregulated by cipargamin and (+)-SJ733. We did this by loading Dd2-Polδ-PfATP4$^{G358S}$ and parental Dd2-Polδ parasites with the Na$^+$-sensitive fluorescent dye SBFI, and measuring [Na$^+$]$_{cyt}$ in parasites exposed to a range of concentrations of cipargamin and (+)-SJ733. We found that Dd2-Polδ-PfATP4$^{G358S}$ parasites were highly resistant to having their [Na$^+$]$_{cyt}$ perturbed by both cipargamin and (+)-SJ733 (Fig. 3a, b). It was not possible to determine precise IC$_{50}$ values for cipargamin- and (+)-SJ733-mediated Na$^+$ dysregulation in Dd2-Polδ-PfATP4$^{G358S}$ parasites due to the very high concentrations that would have been required to obtain a maximal effect. Nevertheless, it is estimated that the IC$_{50}$ values for [Na$^+$]$_{cyt}$ dysregulation would be > 230-fold higher for cipargamin, and > 180-fold higher for (+)-SJ733, in Dd2-Polδ-PfATP4$^{G358S}$ parasites than in the parental Dd2-Polδ parasites (Table 1).

We also investigated the effects of cipargamin and (+)-SJ733 on Na$^+$ regulation in the related apicomplexan parasite *T. gondii*. We used CRISPR-Cas9 to generate *T. gondii* parasites (TgATP4$^{G419S}$-HA) that harbour the equivalent G419S mutation in TgATP4 (Supplementary Fig. 3). We isolated successfully edited clones (Supplementary Fig. 4)

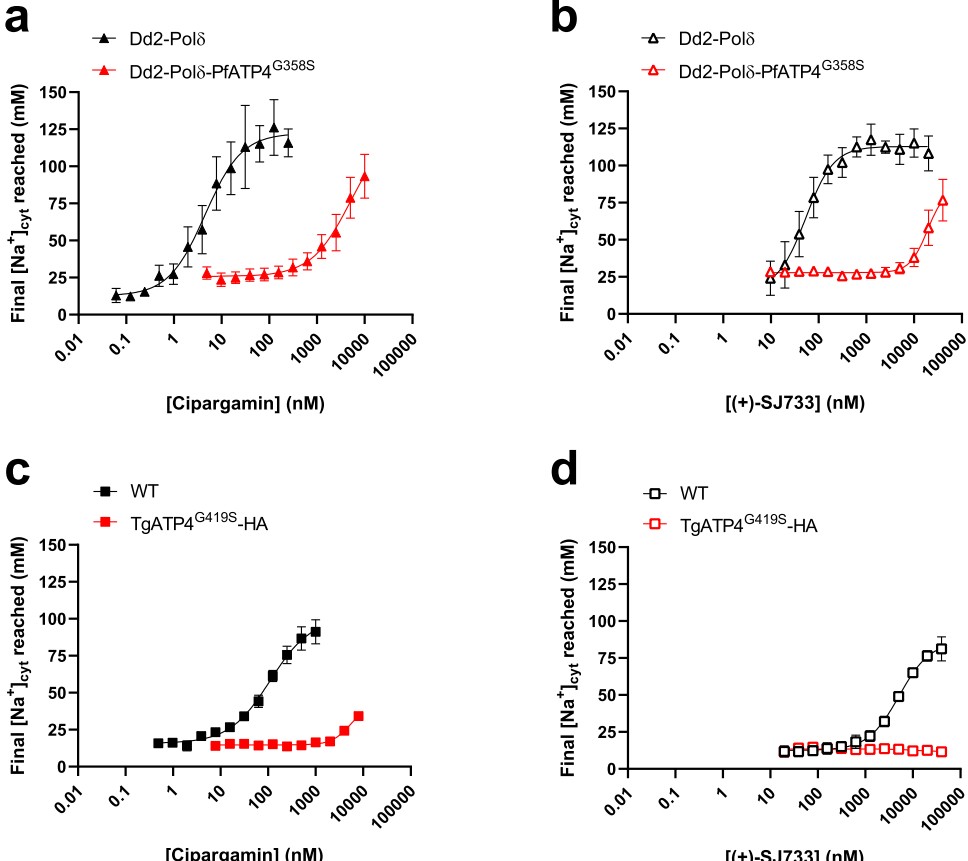

**Fig. 3 | Parasites with the G358S mutation in PfATP4 or the G419S mutation in TgATP4 are resistant to cipargamin- and (+)-SJ733-mediated Na⁺ dysregulation.** The final $[Na^+]_{cyt}$ reached after a ~90 min exposure to a range of concentrations of cipargamin (**a**, **c**) or (+)-SJ733 (**b**, **d**). The measurements were performed at 37 °C in pH 7.1 Physiological Saline Solution with isolated SBFI-loaded Dd2-Polδ (black) and Dd2-Polδ-PfATP4$^{G358S}$ (red) *P. falciparum* trophozoites (triangles; **a**, **b**) and with extracellular SBFI-loaded *T. gondii* tachyzoites (squares; **c**, **d**) expressing TgATP4$^{WT}$ (black) or TgATP4$^{G419S}$-HA (red). The data are the mean (± SEM) from the following

number of independent experiments (performed on different days): 5 (**a**, Dd2-Polδ; with each concentration tested 3–5 times), 6 (**a**, Dd2-Polδ-PfATP4$^{G358S}$; with each concentration tested 3-6 times), 4 (**b**, Dd2-Polδ; all concentrations), 4 (**b**, Dd2-Polδ-PfATP4$^{G358S}$; with each concentration tested 3-4 times), 3 (**c**; for both lines and all concentrations), 4 (**d**, WT; with all concentrations tested 4 times except for the highest concentration which was tested 3 times), 3 (**d**, TgATP4$^{G419S}$-HA; all concentrations). Where not shown, error bars fall within the symbols. Source data are provided as a Source Data file.

**Table 1 | Susceptibility of Dd2-Polδ and Dd2-Polδ-PfATP4$^{G358S}$ *P. falciparum* parasites and TgATP4$^{WT}$ and TgATP4$^{G419S}$-HA *T. gondii* parasites to Na⁺ dysregulation and inhibition of ATP4-associated ATPase activity by cipargamin and (+)-SJ733**

| | IC₅₀ for Na⁺ dysregulation (nM) | | IC₅₀ for ATP4-associated ATPase activity (nM) | |
|---|---|---|---|---|
| | **Cipargamin** | **(+)-SJ733** | **Cipargamin** | **(+)-SJ733** |
| Dd2-Polδ | 8.4 ± 3.0 (5) | 55 ± 15 (4) | 2.5 ± 0.8 (4) | 5.2 ± 2.3 (3) |
| Dd2-Polδ-PfATP4$^{G358S}$ | >2000 (6) | >10,000 (4) | 99 ± 21 (4) (***P = 0.004***) | 829 ± 25 (3) (***P = 5 × 10⁻⁶***) |
| WT *T. gondii* (TgATP4$^{WT}$) | 110 ± 3 (3) | 5045 ± 572 (4) | 8.4 ± 5.8 (4) | 17.8 ± 9.2 (3) |
| TgATP4$^{G419S}$-HA | >8000 (3) | >40,000 (3) | 288 ± 111 (4) (***P = 0.05***) | > 4000 (3) |

The IC₅₀ values (mean ± SEM) are shown, with the number of independent experiments (performed on different days) shown in brackets. For each compound, assay type and parasite species, the IC₅₀ values (where possible to obtain) for the ATP4 mutant and its WT counterpart were compared using two-tailed unpaired *t* tests. *P* values ≤ 0.05 indicate statistical significance and are shown in bold. Source data are provided as a Source Data file.

and confirmed that the parasites expressed TgATP4$^{G419S}$-HA at the plasma membrane (Supplementary Fig. 5). In contrast to *P. falciparum*, *T. gondii* does not require TgATP4 activity for proliferation[18], and the much higher concentrations of cipargamin needed to kill *T. gondii* parasites[42] likely do so via off target effects. However, TgATP4 is required for the maintenance of a low $[Na^+]_{cyt}$ in extracellular tachyzoites under physiological (~130 mM extracellular Na⁺) conditions[18]. We investigated the potency with which cipargamin and (+)-SJ733 dysregulate $[Na^+]_{cyt}$ in extracellular *T. gondii* tachyzoites (TATi/Δku80 strain) expressing wild-type TgATP4 (TgATP4$^{WT}$) and TgATP4$^{G419S}$-HA.

These assays revealed that TgATP4$^{G419S}$-HA parasites were much less sensitive to having their $[Na^+]_{cyt}$ dysregulated by cipargamin and (+)-SJ733 than TgATP4$^{WT}$ parasites (Fig. 3c, d; Table 1).

To date, the most direct assay available for measuring PfATP4 activity entails measuring Na⁺-dependent ATPase activity in *P. falciparum* membrane preparations[11,15]. Approximately 25% of the total ATPase activity measured in *P. falciparum* membrane preparations is Na⁺-dependent. Through studies with PfATP4 inhibitors and *pfatp4* mutant lines, this Na⁺-dependent ATPase activity has been attributed to PfATP4[11,15]. We adapted the ATPase assay for use with *T. gondii* and

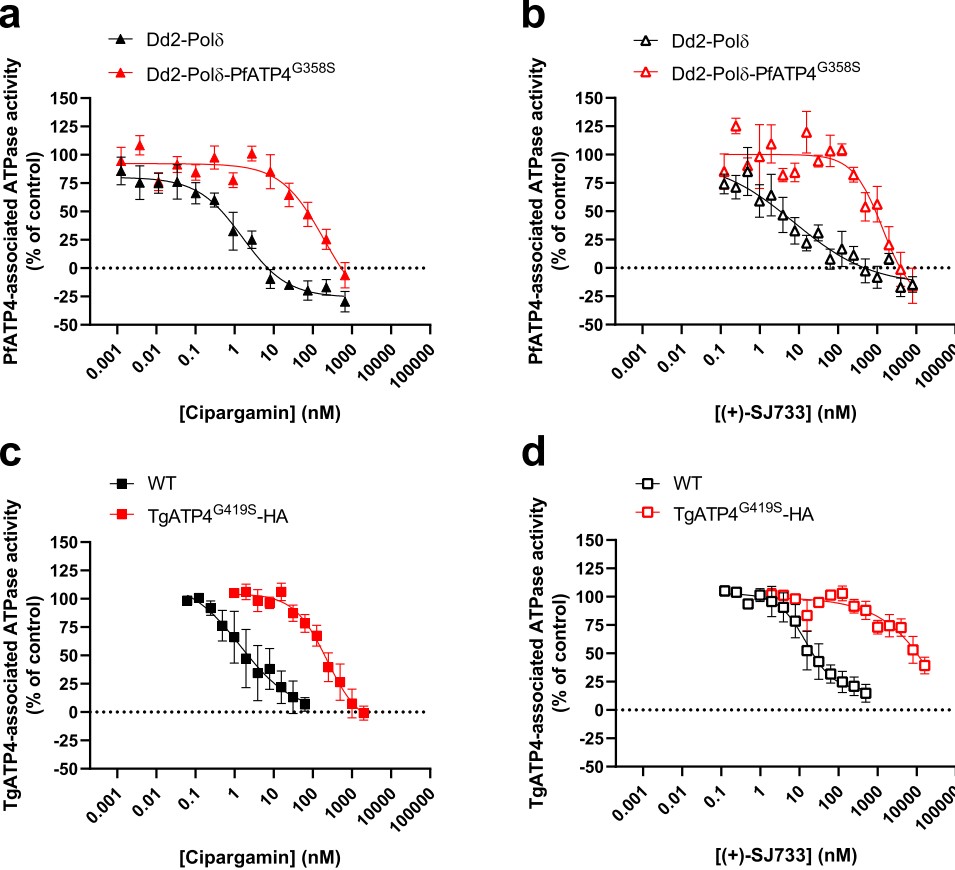

**Fig. 4 | Reduced sensitivity of PfATP4$^{G358S}$- and TgATP4$^{G419S}$-associated ATPase activity to inhibition by cipargamin and (+)-SJ733.** The potency by which cipargamin (**a**, **c**) and (+)-SJ733 (**b**, **d**) inhibit ATP4-associated ATPase activity was determined using membranes prepared from Dd2-Polδ (black) and Dd2-Polδ-PfATP4$^{G358S}$ (red) *P. falciparum* trophozoites (triangles; **a**, **b**) or *T. gondii* tachyzoites (squares; **c**, **d**) expressing TgATP4$^{WT}$ (black) or TgATP4$^{G419S}$-HA (red). The data are

the mean (± SEM) from the following number of independent experiments (each performed on different days with different membrane preparations): 4 (**a**), 3 (**b**), 4 (**c**; with the exception of the lowest cipargamin concentration, which is the mean ± range/2 from $n = 2$ for WT and the mean ± SEM from $n = 3$ for TgATP4$^{G419S}$-HA) and 3 (**d**). Where not shown, error bars fall within the symbols. Source data are provided as a Source Data file.

detected a cipargamin-sensitive Na$^+$-dependent ATPase activity in *T. gondii* membrane preparations (Fig. 4c, d).

We investigated whether the G358S mutation in PfATP4, and the G419S mutation in TgATP4, rendered their ATPase activity less sensitive to inhibition by cipargamin and (+)-SJ733. Using membranes prepared from Dd2-Polδ-PfATP4$^{G358S}$ and their parental Dd2-Polδ parasites, we found that PfATP4$^{G358S}$-associated ATPase activity was 40-fold and 159-fold less sensitive to inhibition by cipargamin and (+)-SJ733, respectively, compared with PfATP4$^{WT}$ (Fig. 4a, b; Table 1). A similar result was obtained with membranes prepared from TgATP4$^{G419S}$-HA parasites and WT parasites, with TgATP4$^{G419S}$-HA-associated ATPase activity displaying a 34-fold and >225-fold lower susceptibility to cipargamin and (+)-SJ733, respectively, than for TgATP4$^{WT}$ (Fig. 4c, d; Table 1). These results provide strong evidence that the G358S mutation in PfATP4, and the equivalent G419S mutation in TgATP4, greatly reduce the sensitivity of the proteins to inhibition by cipargamin and (+)-SJ733.

To investigate how the G358S mutation reduces the sensitivity of PfATP4 to inhibition by cipargamin and (+)-SJ733, we explored the likely binding sites of these compounds in PfATP4 using molecular docking with AutoDock Vina[43]. As no atomic resolution structures are available for PfATP4 we utilised two structures generated using ColabFold[44] (which utilises AlphaFold2[45]), one created by us and one contributed to Open Source Malaria by the Ersilia Open Source Initiative (https://github.com/ersilia-os/osm-pfatp4-structure). The resulting structures have only minor differences, primarily in side-

chain positions. The root mean square deviation of the protein back-bones was 4.5 Å, and of the sidechains 6.5 Å. Blind docking across the entire protein surface showed multiple candidate binding sites of similar affinity for both cipargamin and (+)-SJ733 (Supplementary Fig. 6), including one located in a cavity beside G358, a region where known inhibitors (thapsigargin and cyclopiazonic acid) bind in a different type II P-type ATPase, SERCA[46,47]. Focused docking on this region showed that cipargamin binds in close proximity to G358 (Fig. 5a). Introducing the G358S mutation in either structure reduced the predicted binding affinity and shifted the highest ranked pose away from the site of mutation (Fig. 5a, Supplementary Table 6). This appeared to be due to steric clashes between the serine side chain and cipargamin that persisted even if we allowed flexibility to this residue in our docking protocol. A similar high affinity site was seen for (+)-SJ733 in the Open Source structure (Fig. 5b) that was removed by the G358S mutation due to steric clashes. In our own PfATP4 model, (+)-SJ733 did not bind with similar high affinity to either WT PfATP4 or the G358S variant. While the lack of a high-resolution protein structure for PfATP4 makes determining the exact binding site challenging, these data provide a plausible mechanism by which the affinity for PfATP4 of both cipargamin and (+)-SJ733 is reduced by the introduction of steric clashes with the S358 side chain.

**Effect of the G358S mutation on PfATP4 function**
An important question to address was whether the G358S mutation in PfATP4 impairs the protein's function, and whether or not this leads to

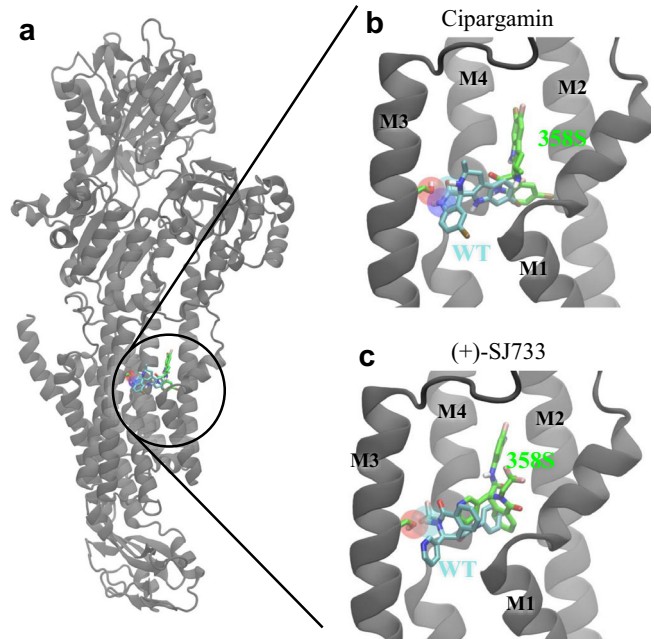

**Fig. 5 | Binding sites proximal to PfATP4 residue 358 predicted by molecular docking to the Open Source structure. a** The location of the lowest energy poses found for cipargamin docked against WT (cyan) and G358S mutant (green) PfATP4, in relation to the entire protein's structure. The side chain of S358 is also shown in green at its position on M3. Close ups of the binding locations of (**b**) cipargamin and (**c**) (+)-SJ733 are shown, with the compounds' carbon atoms in the lowest energy pose shown in cyan for WT PfATP4 and with green for G358S mutant PfATP4. The space occupied by the atoms in the WT binding pose that sterically clash with the S358 side chain are highlighted by transparent balls.

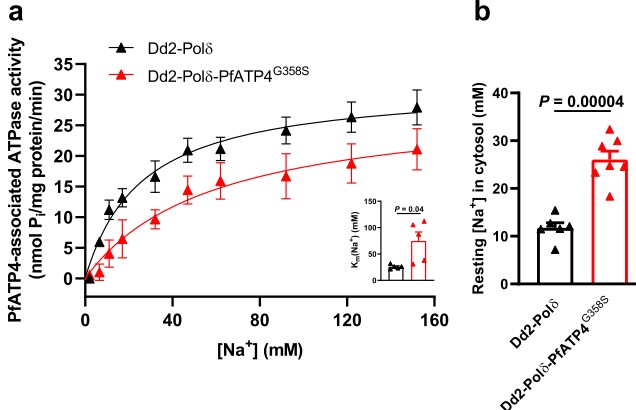

**Fig. 6 | The G358S mutation in PfATP4 affects the Na⁺-dependence of PfATP4-associated ATPase activity. a** Effect of [Na⁺] on PfATP4-associated ATPase activity in membranes prepared from Dd2-Polδ-PfATP4$^{G358S}$ parasites (red triangles) and their Dd2-Polδ parents (black triangles). The data shown are the mean (± SEM) from five independent experiments, each performed on different days with different membrane preparations. The Michaelis–Menten equation (PfATP4-associated ATPase activity = $V_{max}$ × [Na⁺]/([Na⁺] + $K_m$(Na⁺))) was fitted to the data. The $K_m$(Na⁺) in the two lines is shown in the inset, with the symbols showing the $K_m$(Na⁺) values from each individual experiment, the bars showing the mean, and the error bars showing the SEM. A two-tailed paired $t$ test was used to compare the $K_m$(Na⁺) values obtained for the two lines. **b** Resting cytosolic [Na⁺] in Dd2-Polδ-PfATP4$^{G358S}$ parasites (red) and their Dd2-Polδ parents (black). The measurements were performed with isolated trophozoite-stage parasites loaded with the Na⁺-sensitive dye SBFI, and suspended in pH 7.1 Physiological Saline Solution at 37 °C. The bars show the mean + SEM obtained from six (Dd2-Polδ) or seven (Dd2-Polδ-PfATP4$^{G358S}$) independent experiments for each line (performed on different days), and the symbols show the results obtained in individual experiments. The $P$ value is from a two-tailed unpaired $t$ test. $P$ values ≤ 0.05 indicate statistical significance and are shown in bold. Source data are provided as a Source Data file.

a defect in parasite growth. Membrane ATPase assays have been used previously to compare the biochemical properties of Dd2-PfATP4 and Dd2-PfATP4$^{T418N,P990R}$, revealing that the T418N and P990R mutations are associated with a slight decrease in the affinity of PfATP4 for Na⁺ (a 1.3-fold elevation in $K_m$(Na⁺)), but no change in its maximum rate ($V_{max}$)[15]. We investigated the Na⁺-dependence of PfATP4-associated membrane ATPase activity in membranes prepared from Dd2-Polδ and Dd2-Polδ-PfATP4$^{G358S}$ parasites (Fig. 6a). With membranes prepared from Dd2-Polδ-PfATP4$^{G358S}$ parasites, we estimated a $K_m$(Na⁺) for PfATP4$^{G358S}$ of 75 ± 17 mM (mean ± SEM, $n$ = 5; Fig. 6a). This was significantly higher than the $K_m$(Na⁺) estimated for WT PfATP4 in the parental Dd2-Polδ parasites, which was 25.8 ± 1.8 mM (mean ± SEM, $n$ = 5; $P$ = 0.04, two-tailed paired $t$ test; Fig. 6a). The $V_{max}$ values estimated for WT PfATP4 and PfATP4$^{G358S}$ were not significantly different from one another (31.7 ± 3.0 and 30.5 ± 3.9 nmol Pi per mg of (total) protein per min, respectively; mean ± SEM, $n$ = 5; $P$ = 0.6, two-tailed paired $t$ test).

Consistent with the finding of a reduced affinity of PfATP4$^{G358S}$ for Na⁺, we found that Dd2-Polδ-PfATP4$^{G358S}$ parasites had a significantly (2.2 fold) higher resting [Na⁺]$_{cyt}$ than their Dd2-Polδ parents (Fig. 6b). The resting [Na⁺]$_{cyt}$ concentrations (mean ± SEM) for the Dd2-Polδ and Dd2-Polδ-PfATP4$^{G358S}$ parasites were 11.7 ± 1.1 mM ($n$ = 6) and 26.0 ± 1.8 mM ($n$ = 7), respectively.

Of the six PfATP4-mutant parasite lines for which resting [Na⁺]$_{cyt}$ values have been reported previously, five (PfATP4$^{A211T}$, PfATP4$^{I203L/P990R}$, PfATP4$^{L350H}$, PfATP4$^{I398F/P990R}$ and PfATP4$^{T418N,P990R}$) were found to have an elevated resting [Na⁺]$_{cyt}$ compared to their parents[11,20,23]. We confirmed that the Dd2-PfATP4$^{T418N,P990R}$ parent of the HCR clones had a higher [Na⁺]$_{cyt}$ than its Dd2 parent (Supplementary Fig. 7). For HCR1 and HCR2 parasites, the mean [Na⁺]$_{cyt}$ was slightly higher than that of their Dd2-PfATP4$^{T418N,P990R}$ parent, but this was not statistically significant (Supplementary Fig. 7).

To investigate further how frequently mutations in PfATP4 are associated with an elevation of the parasite's resting [Na⁺]$_{cyt}$ we tested an additional five PfATP4-mutant lines that were selected previously with various PfATP4 inhibitors (PfATP4$^{T418N}$, PfATP4$^{Q172K}$, PfATP4$^{A353E}$, PfATP4$^{P966S}$ and PfATP4$^{P966T}$). Four of the five lines had a significantly ($P$ < 0.05) higher resting [Na⁺]$_{cyt}$ than their parents (Supplementary Fig. 7). Thus, considering all the lines expressing a single mutant variant of *pfatp4* for which resting [Na⁺]$_{cyt}$ measurements have now been performed, 10 of the 12 mutant variants of PfATP4, including PfATP4$^{G358S}$, have been associated with an elevation of resting [Na⁺]$_{cyt}$.

## Effect of the G358S mutation in PfATP4 on the growth of asexual blood-stage parasites

To determine whether the G358S mutation in PfATP4 was associated with a cost to the fitness of asexual blood-stage parasites, we investigated the in vitro growth of Dd2-Polδ-PfATP4$^{G358S}$ parasites and their Dd2-Polδ parents in the absence of any drug pressure (Fig. 7). The intraerythrocytic parasite increases in volume as it matures into a young trophozoite, mature trophozoite and then a schizont. In general, a change in growth rate can result from either a decrease in the number of viable parasites produced when a schizont segments to form daughter parasites each cycle, or from a lengthening of the (normally ~48 h) cycle. To test for these possibilities, we: (1) adjusted parasite cultures to a parasitaemia of 1% and measured the parasitaemias after one complete cycle; and (2) measured the change in volume distribution of trophozoites after intervals of ~48 h.

To compare the number of viable parasites produced per cycle by Dd2-Polδ-PfATP4$^{G358S}$ parasites and their Dd2-Polδ parents, we used flow cytometry to determine the parasitaemia in three replicate cultures containing synchronous trophozoite-stage parasites, then

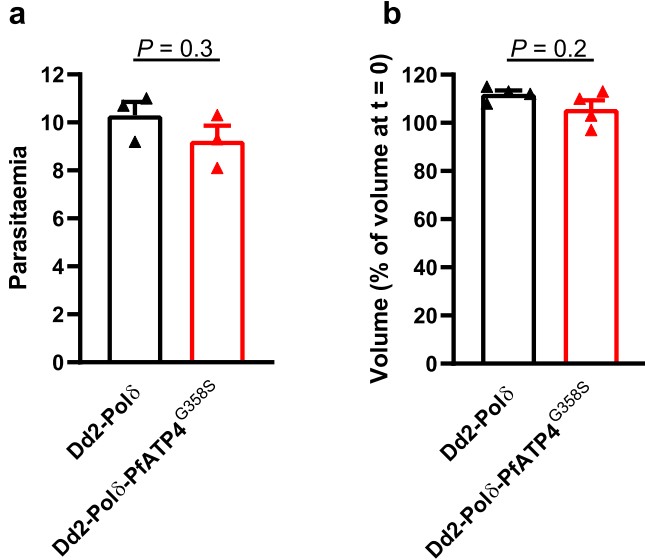

**Fig. 7 | Production of viable daughter parasites and progression through the erythrocytic cycle is similar in Dd2-Polδ-PfATP4$^{G358S}$ (red) and Dd2-Polδ (black) parasites. a** The parasitaemias obtained one cycle (~48 h) after adjusting the parasitaemia of cultures to 1%. The bars show the mean from three independent experiments, each performed on different days with three cultures (technical replicates). **b** Parasite volume (as a percentage of that measured at 0 h) after one cycle (48 h) of growth. The bars show the mean from four independent experiments, each performed on different days with at least two technical replicates. In (**a**) and (**b**), the error bars are SEM, and the symbols show the data from each independent experiment. The *P* values are from two-tailed unpaired *t* tests. Source data are provided as a Source Data file.

diluted each one to a parasitaemia of 1%. The parasitaemias in each culture were then determined approximately 48 h later using flow cytometry. This was performed on three independent occasions, yielding the data shown in Fig. 7a. One cycle (~48 h) after being adjusted to a parasitaemia of 1%, the parasitaemia reached $10.3 \pm 0.5\%$ in the Dd2-Polδ cultures, and $9.2 \pm 0.6\%$ in the Dd2-Polδ-PfATP4$^{G358S}$ cultures (mean ± SEM, $n = 3$; $P = 0.3$, two-tailed unpaired *t* test), consistent with Dd2-Polδ and Dd2-Polδ-PfATP4$^{G358S}$ parasites producing a similar number of viable daughter parasites per cycle.

To compare the rate of progression through the asexual blood-stage cycle in Dd2-Polδ and Dd2-Polδ-PfATP4$^{G358S}$ parasites, we measured the mean volume of isolated trophozoites, then determined the change in mean volume 48 h later (Fig. 7b). As a percentage of the starting volume, the mean trophozoite volume after 48 h was $112 \pm 1\%$ for Dd2-Polδ parasites, and $106 \pm 4\%$ for the Dd2-Polδ-PfATP4$^{G358S}$ parasites (mean ± SEM, $n = 4$; $P = 0.2$, two-tailed unpaired *t* test). The small increase in mean parasite volume between the 0 h and 48 h time points observed for both lines suggests that the duration of their erythrocytic life cycles is slightly shorter than 48 h.

Thus, despite affecting the function of PfATP4, the G358S mutation was not associated with a significant change in the length of the intra-erythrocytic cycle or in the number of viable merozoites produced per cycle during the asexual blood stage. This is consistent with our observation that there was no obvious difference in the growth of these parasites during the routine culture of the lines.

### Effect of the G358S mutation in PfATP4 on parasite transmission and development in mosquitoes

Lastly, we investigated whether the G358S mutation in PfATP4 influenced the ability of parasites to infect mosquitoes and develop within them. These experiments were performed with parasites generated from the transmissible NF54 strain: NF54$^{WT}$ (parent), NF54$^{CTL}$ (control

line with silent binding-site mutations), and three NF54-PfATP4$^{G358S}$ lines (denoted 1, 2 and 3). Gametocytes were generated in vitro and *Anopheles stephensi* mosquitoes were allowed to feed on the gametocyte cultures. On Day 12 post blood meal, the mosquitoes were examined to determine the prevalence and intensity of infection. For all three NF54-PfATP4$^{G358S}$ lines, the NF54$^{CTL}$ line and the NF54 parental line, the prevalence of infection was between 90–100% (i.e. 90–100% of the mosquitoes that fed on the gametocyte cultures became infected). The intensity of infection (i.e. number of oocysts in the midguts of infected mosquitoes) is shown in Fig. 8a. While the infection intensities for the NF54$^{G358S}$ lines and the NF54$^{CTL}$ line were somewhat lower than those for the NF54$^{WT}$ line, parasites from each of the lines formed healthy-appearing oocysts (Supplementary Fig. 8), indicating that the G358S mutation in PfATP4 did not affect the establishment of infection in mosquitoes.

On Day 16 post blood meal, mosquitoes infected with the different parasite lines were checked for salivary gland sporozoites (Fig. 8b). Salivary gland sporozoite loads were somewhat lower in mosquitoes infected with the NF54$^{CTL}$ and NF54$^{G358S}$ parasites compared to those infected with NF54$^{WT}$ parasites, with salivary gland sporozoite loads for the NF54$^{G358S}$ lines ranging from an average of 37,500 sporozoites to 47,304 sporozoites per mosquito. Nonetheless, the G358S mutation in PfATP4 did not substantially hinder the ability of the parasite to be transmitted through mosquitoes.

Edited parasite genomes were analyzed by whole-genome sequencing and compared to that of the NF54$^{WT}$ parental line to search for genetic changes that might have spontaneously arisen during prolonged in vitro culture and associate with altered transmissibility to mosquitoes (Supplementary Tables 2–4). We confirmed the presence of the PfATP4 G358S mutation in 100% of the parasite population in the two gene-edited mutant lines, NF54$^{G358S-1}$ and NF54$^{G358S-2}$, along with the presence of 17 silent binding-site mutations at the *pfatp4* guide RNA cleavage site that were introduced as part of the CRISPR-Cas9 editing strategy. These silent mutations were also present in the edited control line NF54$^{CTL}$ that expresses wild-type PfATP4 (Supplementary Tables 3, 4). We also identified two additional high confidence SNPs coding for non-synonymous mutations that were absent in the NF54$^{WT}$ parental strain: (1) a SNP in PF3D7_0304000, coding for a V78A change in inner membrane complex protein 1a, present in both NF54$^{CTL}$ and the PfATP4-mutant line NF54$^{G358S-2}$; and (2) a SNP in PF3D7_1251500, coding for a N521Y change in the ATP-dependent RNA helicase DRS1, found only in NF54$^{G358S-1}$ (Supplementary Tables 3, 4). No common SNP was found in the *pfatp4*-edited lines that could explain the slightly lowered infection intensities and sporozoite loads compared with NF54$^{WT}$ parasites. We also observed no CNVs in any of the edited parasite lines.

## Discussion

PfATP4 has emerged as a major antimalarial drug target, with the PfATP4 inhibitor cipargamin having undergone extensive testing in Phase 1 and Phase 2 clinical trials[27,40], and another PfATP4 inhibitor (+)-SJ733 also having been tested in humans[26]. Understanding the likelihood and mechanisms of resistance to PfATP4 inhibitors is important as compounds with this mechanism are pursued further. In this study we generated highly cipargamin-resistant parasites in three independent selections with cipargamin: two stepwise selections commencing with low-level resistant parasites, and one single-step selection commencing with parasites displaying a hypermutator phenotype. In each case these selections gave rise to parasites with a G358S mutation in PfATP4. A fourth independent experiment performed with Dd2-Polδ in a different laboratory also yielded the same result[41]. In contrast, many different mutations in PfATP4 have been reported in parasites selected under conditions in which the acquisition of a low level of resistance to cipargamin is sufficient for

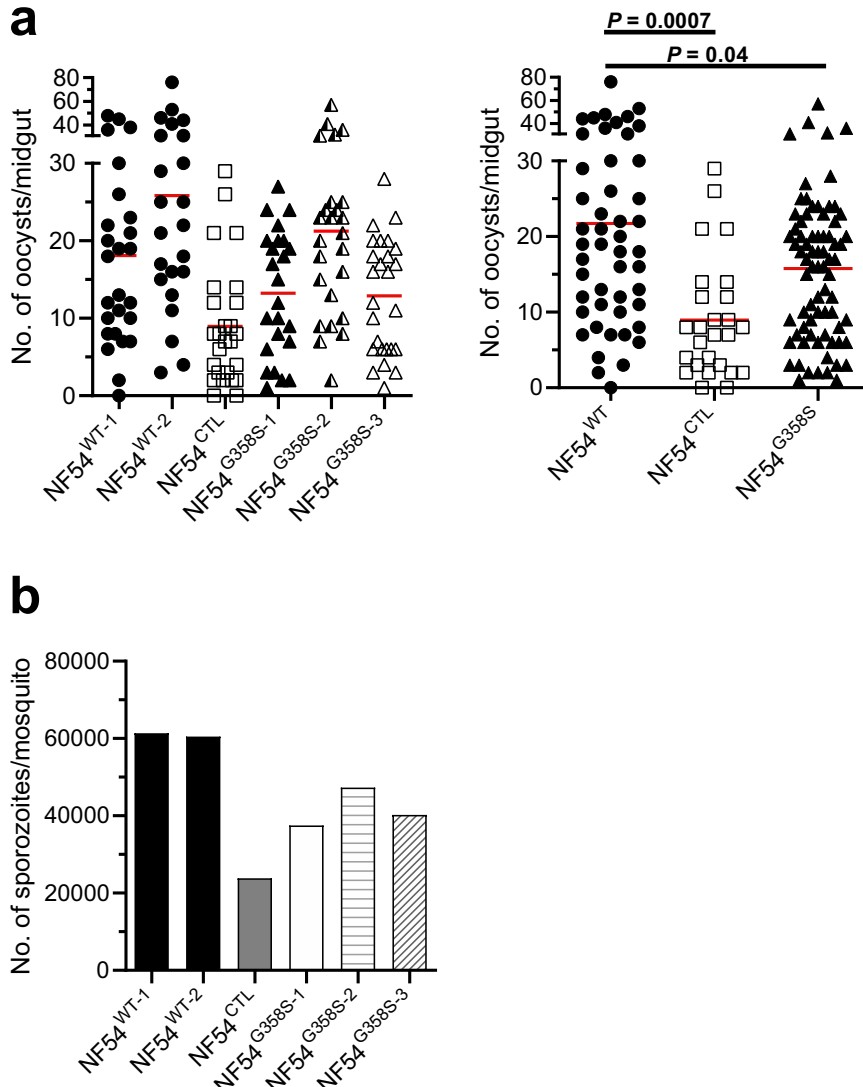

**Fig. 8 | Parasites with the G358S mutation in PfATP4 are infective to mosquitoes and produce salivary gland sporozoites. a** Left Panel: Oocyst numbers in the midguts of *Anopheles stephensi* mosquitoes fed on gametocyte cultures of the parasite lines indicated ($n = 23$–26 mosquitoes per line; determined on Day 12 post blood meal). The red bar indicates the mean. Data for two NF54[WT] cultures (prepared from separate vials that had been frozen on different dates; closed circles) are shown. Right Panel: Pooled data for statistical analysis. Because of the high variability inherent in these data, we pooled the two NF54[WT] controls (closed circles) and the three mutant lines (NF54[G358S-1], [-2], and [-3]; closed triangles) for analysis.

There was a small difference in oocyst numbers between NF54[WT] and NF54[G358S] ($P = 0.04$; Mann–Whitney test, two-tailed). There was a larger difference in oocyst numbers between NF54[WT] and NF54[CTL] (open squares; $P = 0.0007$; Mann–Whitney test, two-tailed). $P$ values $\leq 0.05$ indicate statistical significance and are shown in bold. **b** Numbers of salivary gland sporozoites, determined on Day 16 post blood meal. The data are the mean number of sporozoites per mosquito. For each parasite line, 20 mosquitoes were dissected, their salivary glands pooled and homogenised, and sporozoites counted using a haemocytometer. Source data are provided as a Source Data file.

survival[9,35]. Thus, it would appear that there are many roads to low-level cipargamin resistance but few roads to high-level cipargamin resistance. The frequency of resistance, measured at ~2 × 10$^{-8}$ to 1 × 10$^{-7}$ for low-level cipargamin resistance[35], decreased to <6.6 × 10$^{-9}$ for high-level resistance (when starting with Dd2-PfATP4[T418N,P990R] parasites). Indeed, we only succeeded in generating highly resistant parasites via a single-step procedure using genetically engineered parasites with mutations in DNA polymerase δ and a hypermutator phenotype. These parasites have a mutation rate ~5-8-fold higher than wild-type Dd2 parasites in coding regions of the genome under normal conditions, and 13-28 fold higher when parasites are exposed to antiplasmodial compounds[41].

The G358S mutation in PfATP4 was observed in 22/25 instances of recrudescence in a recent clinical trial for cipargamin[40]. On five of these occasions there was a mix of G358S with wild-type sequence at

the positions tested, G358A or G359A. There were also three instances of recrudescence in which the G358S mutation was not observed. These were associated (on one occasion each) with the G359A, L354V and L181F mutations in PfATP4[40]. The results of this trial demonstrate the clinical significance of the G358S mutation and provide further evidence that it may be one of few mutations that can confer high-level resistance to cipargamin. It will be interesting to investigate the G358A, G359A, L354V and L181F mutations in future studies to assess the level of resistance conferred by those mutations and their effects on PfATP4 function and parasite fitness.

In contrast to cipargamin, for which high-level resistance had not been observed previously in in vitro evolution studies, high-level resistance to SJ733 has been encountered before and there appear to be multiple mutations that can confer this phenotype[23,48]. Genetically engineered parasites bearing the L350H or P412T mutations in PfATP4

have been shown to display a high level of resistance to SJ733, with IC$_{50}$s of 3.5 μM and 10 μM, respectively, compared to 75 nM for parental wild-type parasites[48]. All the parasite lines with a G358S mutation in PfATP4 tested in this study had IC$_{50}$s for (+)-SJ733 > 9 μM.

The HCR parasites generated in this study have the highest level of PA21A050 resistance reported to date (mean IC$_{50}$s of ~60 nM). Previous selections with a pyrazoleamide yielded parasites with an IC$_{50}$ for PA21A050 of 16 nM (compared to 0.7 nM for the parental line) and with mutations in five proteins, including PfATP4 (V178I) and the Ca$^{2+}$-dependent protein kinase PfCDPK5 (T392A)[25]. Whole-genome sequencing of our HCR parasites did not reveal any mutations in the gene encoding PfCDPK5 (or in the three other genes that were mutated along with *pfatp4* in the pyrazoleamide-selected lines) relative to their parents.

The HCR parasites were more resistant to cipargamin, (+)-SJ733 and PA21A050 than the Dd2-Polδ-PfATP4$^{G358S}$ parasites, and were more resistant to cipargamin and (+)-SJ733 than the NF54$^{G358S}$ parasites (PA21A050 was not tested against these parasites). The two mutations in Dd2-PfATP4$^{T418N,P990R}$ parasites conferred some resistance to each drug (ranging from 2-14-fold), with the fold difference in IC$_{50}$ value between HCR parasites and their Dd2 'grandparents' equating to approximately the product of the fold difference between Dd2 and Dd2-PfATP4$^{T418N,P990R}$ parasites (resulting from the T418N and P990R mutations in PfATP4) and that between Dd2-PfATP4$^{T418N,P990R}$ and HCR parasites (stemming from the amplification of *pfatp4* and the presence of the G358S-coding mutation in one of the copies). The G358S mutation in PfATP4 was an important contributor to high-level resistance for cipargamin and (+)-SJ733, with all parasites with this mutation having IC$_{50}$ values > 750-fold and > 138-fold higher than their PfATP4$^{WT}$ counterparts, respectively. The contribution of the G358S mutation was less pronounced for PA21A050, with the Dd2-Polδ-PfATP4$^{G358S}$ parasites displaying only a 5-fold increase in IC$_{50}$ (c.f. a ~50-fold difference between HCR parasites and Dd2).

Relative to their Dd2-PfATP4$^{T418N,P990R}$ parent, HCR1 and HCR2 parasites were both more sensitive to dihydroartemisinin (both with ~1.6-fold lower IC$_{50}$s), and HCR2 was less susceptible to chloroquine (1.3-fold higher IC$_{50}$). The reasons for these differences are not clear, although for HCR2 parasites, the reduction in *pfmdr1* copy number relative to HCR1 and Dd2-PfATP4$^{T418N,P990R}$ parasites might play a role[49]. Our assays with other cipargamin-selected and genetically modified lines revealed no effect of the G358S mutation on parasite susceptibility to dihydroartemisinin, pyronaridine, piperaquine, monodesethyl-amodiaquine, lumefantrine, KAF156 or INE963[34], showing that resistance to cipargamin associated with the PfATP4 G358S mutation could be addressed with adequate drug combinations.

Our studies revealed that the G358S mutation in PfATP4, and the G419S mutation in TgATP4, rendered the Na$^+$-ATPase activity of the proteins resistant to inhibition by cipargamin and (+)-SJ733. Molecular docking studies with two ColabFold models of PfATP4 provided a possible mechanistic explanation, suggesting that the G358S mutation creates a steric clash that reduces the binding affinity of cipargamin and (+)-SJ733. TgATP4$^{WT}$-associated ATPase activity was about 3-fold less sensitive to inhibition by both cipargamin and (+)-SJ733 than PfATP4$^{WT}$-associated ATPase activity. (+)-SJ733 was only ~2-fold less potent than cipargamin against both PfATP4$^{WT}$ and TgATP4$^{WT}$ (as measured in cell-free assays), yet the difference in potency between the drugs was greater (>6-fold) in [Na$^+$]$_{cyt}$ assays (performed with intact isolated *P. falciparum* or extracellular *T. gondii* parasites) and in parasite proliferation assays with *P. falciparum* parasites. This raises the possibility that the relative intracellular concentration reached for (+)-SJ733 is lower than that for cipargamin.

*T. gondii* proved useful as a model with which to study ATP4 activity in this study. There is homology between TgATP4 and PfATP4, which allowed the residue equivalent to G358 in PfATP4 to be identified unequivocally in TgATP4. Furthermore, the recent characterisation of TgATP4 provided evidence that the protein performs the same role that has been ascribed to PfATP4[18]. *T. gondii* is highly genetically tractable, with clonal TgATP4$^{G419S}$-HA parasites generated within weeks. Furthermore, *T. gondii* parasites can survive and proliferate in the absence of TgATP4 activity, which can be exploited to study mutations in ATP4 that impair its function. Thus, *T. gondii* might serve as a good model for the investigation of a variety of other resistance-conferring mutations in ATP4, as well as to probe other residues that might be important for ATP4 function.

The G358S mutation in PfATP4 affected the function of the protein. Studies of PfATP4-associated ATPase activity revealed that the affinity of PfATP4$^{G358S}$ for Na$^+$ was lower than that of PfATP4$^{WT}$. Consistent with this, Dd2-Polδ-PfATP4$^{G358S}$ parasites had an elevated resting [Na$^+$]$_{cyt}$ compared to their parents. For HCR parasites, the resting [Na$^+$]$_{cyt}$ appeared slightly increased beyond the elevated [Na$^+$]$_{cyt}$ caused by the other two mutations (T418N and P990R) found in these parasites and their Dd2-PfATP4$^{T418N,P990R}$ parents, but this was not significant. In total, in this study and others[11,20,23], 12 parasites having a (single) mutant variant of PfATP4 have now been investigated in Na$^+$ assays, with 10 of these found to have a significantly elevated [Na$^+$]$_{cyt}$.

We did not observe obvious differences between the growth of asexual blood-stage Dd2-Polδ-PfATP4$^{G358S}$ parasites and their parents. Among other PfATP4-mutant lines, two have been reported to have a growth defect, and three have been reported not to[20,23]. From this study and previous studies[20,23], there are now six PfATP4-mutant lines for which both resting [Na$^+$]$_{cyt}$ data and growth data are available. Of the four that were found to have normal growth, three (PfATP4$^{G358S}$, PfATP4$^{A211T}$, PfATP4$^{I203L/P990R}$) were found to have an elevated resting [Na$^+$]$_{cyt}$ and one (PfATP4$^{A187V}$) was found not to. Of two lines reported to have a growth defect, one (PfATP4$^{L350H}$) was found to have a ~2.8-fold elevated resting [Na$^+$]$_{cyt}$[23] and one (PfATP4$^{P996T}$) was found to have a slight (~1.3-fold) elevation of [Na$^+$]$_{cyt}$ that was not statistically significant. Thus, an elevation in resting [Na$^+$]$_{cyt}$ does not consistently give rise to a defect in parasite growth, at least during the asexual blood stage.

Our studies also revealed that parasites with the PfATP4 G358S mutation could produce gametocytes in vitro, infect mosquitoes, and produce oocysts in the mosquito midgut and subsequently sporozoites in the mosquito salivary glands. Thus, if parasites with the G358S mutation in PfATP4 were to emerge in the field, they would be capable of onward transmission. We did not test the fitness or infectivity of the sporozoites produced in this study, and therefore cannot exclude the possibility that these characteristics are affected by the G358S mutation in PfATP4.

In conclusion, it is possible for parasites to acquire high-level resistance to both cipargamin and (+)-SJ733 while maintaining transmissibility and a normal growth rate. Thus, *pfatp4* codon 358 should be monitored closely in the course of cipargamin clinical development. In prioritising PfATP4 inhibitors for further development, it would be useful to attempt to generate high-level resistance to them with Dd2-Polδ parasites, as well as test them against parasites with the G358S mutation in PfATP4. Importantly, a drug combination approach may serve as a suitable risk mitigation strategy for PfATP4 inhibitors, since parasites with the G358S mutation in PfATP4 remained fully susceptible to other antimalarials with unrelated modes of action.

## Methods

### Use of human blood
The use of human blood in this study was approved by the Australian National University Human Research Ethics Committee (Protocol numbers 2011/266 and 2017/351) and the Johns Hopkins School of Medicine Institutional Review Board (Protocol number NA_00019050).

## Antiplasmodial compounds

Cipargamin, KAF156, pyronaridine, piperaquine, monodesethyl-amodiaquine, lumefantrine, and MMV665949 were kindly provided by MMV. (+)-SJ733 was kindly provided by MMV and Prof. R. Kip Guy. PA21A050 was kindly provided by Assoc. Prof. Erkang Fan and Prof. Akhil Vaidya. INE963 was kindly provided by Novartis. MMV006656 and chloroquine were purchased from Princeton BioMolecular Research and Sigma, respectively. DHA was purchased from Sigma and Selleck Chemicals.

## *P. falciparum* culture

*P. falciparum* parasites were cultured in human erythrocytes[50] and were synchronised by sorbitol treatment[51]. NF54-based lines were cultured at 2% haematocrit in human O + erythrocytes (purchased from pooled, anonymous donors at the New York Blood Center) in RPMI 1640 medium (KD Medical) supplemented with 0.21% sodium bicarbonate (Sigma Aldrich), 10 µg/mL gentamicin (Fisher) and 10% O + human serum. Dd2-B2-PfATP4[G358S] and Dd2-B2 parasites were cultured at 3% haematocrit in human O + erythrocytes in RPMI-1640 media, supplemented with 25 mM HEPES (Fisher), 50 mg/L hypoxanthine (Sigma Aldrich), 2 mM L-glutamine (Cambridge Isotope Laboratories, Inc.), 0.21% sodium bicarbonate (Sigma Aldrich), 0.5% (wt/vol) AlbuMAXII (Invitrogen), 7.5% O + human serum and 10 µg/mL gentamicin (Fisher). Cultures were maintained at 37 °C in an atmosphere consisting of 5% $O_2$, 5% $CO_2$ and 90% $N_2$.

For all other parasites, the cultures, which typically had a haematocrit between 2-4%, were maintained with continuous shaking[52] at 37 °C in a low-$O_2$ atmosphere (1% $O_2$, 3% $CO_2$ and 96% $N_2$). The culture medium was RPMI 1640 containing 25 mM HEPES (Gibco) supplemented with 11 mM additional glucose, 0.2 mM hypoxanthine, 20 µg/mL gentamicin sulphate and 3 g/L Albumax II. The blood was provided by Australian Red Cross Lifeblood without disclosing the identities of the donors.

## *P. falciparum* lines

The Dd2-B2-PfATP4[G358S] line was reported previously[23], as were Dd2-PfATP4[T418N,P990R] and its Dd2 parent[9] (kindly provided by E. Winzeler). Dd2-PfATP4[T418N,P990R] was used to generate highly cipargamin-resistant parasites (Fig. 1a), from which clones HCR1 and HCR2 were derived by limiting dilution[53]. Dd2-Polδ is a parasite line engineered to have a mutant form of DNA polymerase δ[41].

Clonal Dd2-Polδ-PfATP4[G358S] parasites were obtained using a FACS-based method. Briefly, erythrocytes infected with late trophozoite-stage parasites were enriched using a Miltenyi Biotec VarioMACS magnet. A FACS Aria III system (Imaging & Cytometry Facility, The John Curtin School of Medical Research, Australian National University) was used to add single cells to individual wells in a 96-well plate containing culture medium and uninfected erythrocytes (2% haematocrit). Cultures were provided with fresh medium and erythrocytes as required. After 18 days, wells containing parasites were identified by assaying for parasite-specific lactate dehydrogenase activity[53].

Resting $[Na^+]_{cyt}$ measurements for additional *P. falciparum* lines are shown in Supplementary Fig. 7. W2-PfATP4[P966S], W2-PfATP4[P966T] and their W2 parent were reported previously[23] and generously provided by R. K. Guy. Dd2-PfATP4[Q172K], Dd2-PfATP4[A353E], Dd2-PfATP4[T418N] and their parent (Dd2 clone 10A) were generated through in vitro evolution experiments performed previously[24]. The Dd2-PfATP4[T418N] clone and its parent were described previously[15]. Clones of Dd2-PfATP4[Q172K] and Dd2-PfATP4[A353E] were obtained from MMV011567-pressured cultures 1 and 2[24] by limiting dilution.

## Generation of *P. falciparum* NF54 parasites with the G358S mutation in PfATP4

The transmission-competent NF54[WT] strain (BEI Resources, MRA-1000) provided by Johns Hopkins University was used for *pfatp4* gene editing. Mutations in *pfatp4* were engineered into NF54 parasites using a previously published "all-in-one" pDC2-based CRISPR-Cas9 plasmid, pDC2-cam-coSpCas9-U6-gRNA-hdhfr[54]. This plasmid contains expression cassettes for *Cas9* (under control of the *calmodulin* promoter) and the selectable marker human dihydrofolate reductase (h*dhfr*) that confers resistance to WR99210 (under the *P. chabaudi dhfr-ts* (PcDT) promoter), as well as cloning sites for the insertion of a gene-specific guide RNA (gRNA) (under a *U6* promoter) and a gene-specific donor template for homology-directed repair (Supplementary Fig. 2). Two *pfatp4* gRNAs (gRNA 1 and 2) were selected using the online tool ChopChop based on their proximity to the mutation of interest (G358S), guanine-cytosine (GC) content, and the absence of poly adenine/thymine (A/T) tracts (http://chopchop.cbu.uib.no). gRNA primers were annealed and cloned into the pDC2 CRISPR-Cas9 vector by In-Fusion cloning at the BbsI restriction enzyme sites (Supplementary Fig. 2). Donor fragments encoding the G358S mutation plus 17 silent shield mutations at the gRNA 1 and 2 cut sites, or silent shield mutations alone (control), were synthesised by Genewiz, then cloned into the gRNA1 or gRNA2 pDC2 CRISPR-Cas9 plasmid by In-Fusion cloning (Takara) at the EcoRI/AatII sites (Supplementary Fig. 2). Final plasmids were prepped for transfection using the NucleoBond® Xtra midi prep kit (Macherey-Nagel).

Parasites were electroporated with purified plasmid DNA[55]. Briefly, a 2.5 mL culture of predominantly ring-stage NF54 parasites (≥ 5%) was washed with 10 mL 1× Cytomix and resuspended in a final volume of 220 µL Cytomix. This mixture was then added to 100 µg of plasmid DNA (also in 220 µL of Cytomix) and electroporated at a voltage of 0.31 kV and a capacitance of 950 µF in a 2 mm gap cuvette using a Gene Pulser (Bio-Rad). For each transfection (mutations and controls), parasites were co-electroporated with gRNA 1 and gRNA 2 plasmids (50 µg of each). Starting one day post transfection, cultures were selected for 6 days with 1 nM WR99210. Following selection, cultures were maintained in complete medium (RPMI-1640 media (KD Medical), supplemented with 0.21% sodium bicarbonate (Sigma Aldrich), 10 µg/mL gentamicin (Fisher), and 10% O + human serum) until recrudescence. Gene editing in recrudescent parasites was assessed by Sanger sequencing of the *pfatp4* locus from blood PCRs (Bioline) of bulk cultures (Supplementary Fig. 2), and subsequently confirmed by whole-genome sequencing (Supplementary Tables 2−4). Parasite populations harbouring recrudescent PfATP4 G358S mutants were pressured with 7.5 nM cipargamin (-3 x $IC_{50}$ of the parental NF54 strain) for 6 days to remove wild-type (non-edited) parasites and enrich for 100% edited populations.

## Isolation of *P. falciparum* parasites from their host erythrocytes

Prior to the preparation of parasite membranes or to measurements of $[Na^+]_{cyt}$ or cell volume, mature trophozoite-stage parasites (approximately 34–40 h post-invasion) were functionally isolated from their host erythrocytes by brief exposure (of cultures at approximately 4% haematocrit) to saponin (0.05% w/v, of which ≥10% was the active agent sapogenin)[56]. The parasites were then washed several times in bicarbonate-free RPMI 1640 supplemented with 11 mM additional glucose, 0.2 mM hypoxanthine and 25 mM HEPES (pH 7.10), and maintained in this medium at a density of $-1 \times 10^7 - 3 \times 10^7$ parasites mL$^{-1}$ at 37 °C until immediately before their use in experiments.

## *P. falciparum* genomic DNA isolation and sequencing

**Drug-selected lines.** Genomic DNA was isolated from saponin-isolated trophozoite-stage parasites using either a QIAGEN DNeasy Plant Mini Kit or a QIAGEN DNeasy Blood and Tissue Kit. The primers used for *pfatp4* amplification and sequencing are shown in Supplementary Table 7.

Whole-genome sequencing of DNA prepared using the TruSeq DNA sample Prep Kit (Illumina) was performed[57] and the resulting fastq

files were aligned using Bowtie 2 (version 2.2.5)[58] with parameters sensitive-local and maxins 1000. Duplicate reads were removed using Picard tools MarkDuplicates (version 2.2.2). Calling of single nucleotide variants (SNVs) and indels was performed with SNVer (version 0.5.3)[59] and VarScan (version 2.4)[60]. Copy number analysis was performed using the R package QDNaseq (version 1.10.0)[61]. Structural variant calling was performed using GRIDSS (version 1.5)[62]. Paired-end reads were assembled and aligned to two reference genomes, Dd2 (assembly ASM14979v1; https://www.ncbi.nlm.nih.gov/assembly/GCA_000149795.1/) and 3D7 (PlasmoDB-29_Pfalciparum3D7; https://plasmodb.org/plasmo/app/downloads/release-29/Pfalciparum3D7/). Mappability, coverage and alignment statistics were better with alignment to the 3D7 reference genome, and this was the version used in the analysis. Coverage of all samples was equal or greater than 130× (Supplementary Note).

**Genetically modified NF54 parasites**. Genomic DNA was extracted from parasites isolated from their host erythrocytes with saponin (0.2% w/v). The cells were then washed twice with PBS and genomic DNA was extracted using the QIAamp DNA Blood Mini Kit (Qiagen).

Whole-genome sequencing was performed using a Nextera Flex DNA library kit and multiplexed on a MiSeq flow cell to generate 300 bp paired-end reads. Sequences were aligned to the *Pf* 3D7 reference genome (PlasmoDB-48_Pfalciparum3D7; https://plasmodb.org/plasmo/app/downloads/release-48/Pfalciparum3D7/fasta/) using the Burrow-Wheeler Alignment (BWA version 0.7.17). PCR duplicates and unmapped reads were filtered out using Samtools (version 1.13) and Picard MarkDuplicates (GATK version 4.2.2). Base quality scores were recalibrated using GATK BaseRecalibrator (GATK version 4.2.2). GATK HaplotypeCaller (GATK version 4.2.2) was used to identify all possible single nucleotide variants (SNVs) in test parasite lines filtered based on quality scores (variant quality as function of depth QD > 1.5, mapping quality > 40, min base quality score > 18, read depth > 5) to obtain high quality single nucleotide polymorphisms (SNPs) that were annotated using SnpEff version 4.3t[63]. Comparative SNP analyses between the gene-edited PfATP4 G358S mutant (NF54$^{G358S-1}$ and NF54$^{G358S-2}$) and control (NF54$^{CTL}$) lines, and the NF54$^{WT}$ parental strain were performed to generate the final list of SNPs (Supplementary Tables 3, 4). BIC-Seq version 1.1.2[64] was used to discover copy number variants (CNVs) against the parental strain using the Bayesian statistical model. SNPs and CNVs were visually inspected and confirmed using Integrative Genome Viewer (IGV). All gene annotations in the analysis were based on PlasmoDB-48_Pfalciparum3D7 (https://plasmodb.org/plasmo/app/downloads/release-48/Pfalciparum3D7/gff/).

**P. falciparum proliferation assays**
Two types of parasite proliferation assays were employed in this study, with both yielding highly consistent results. For the NF54-based lines and Dd2-B2-PfATP4$^{G358S}$ and its parent, assays were initiated with cultures containing predominantly ring-stage parasites with a parasitaemia of 0.3% and a haematocrit of 1%. All assays using NF54 were performed in culture media containing 10% O + human serum. The duration of the assays was 72 h and the concentration of DMSO was <0.5% (v/v). Serial dilutions of the test compounds were performed using a Tecan D300e digital dispenser, with parasite cultures added manually. Parasite survival was assessed by flow cytometry on an Intellicyt iQue3 (Essen Bioscience), with 1× SYBR Green I (Invitrogen) and 200 nM MitoTracker Deep Red FM (Invitrogen) used to stain the nucleus and the mitochondria in live parasites, respectively[65]. The gating strategy used to quantify parasite survival, performed using FlowJo version 10 (FlowJo LLC.), is illustrated in Supplementary Fig. 9. IC$_{50}$ and IC$_{90}$ values were calculated by linear interpolation of the dose-response data.

For experiments with all other parasites, the effects of compounds of interest on parasite proliferation was measured in 96-well

plates containing serial dilutions of the compounds in culture medium[66], with a fluorescent DNA-intercalating dye used to detect surviving parasites[67]. The assays were initiated with erythrocytes infected with predominantly ring-stage parasites, and the starting parasitaemia and haematocrit were both 1%. The duration of the assays was 72 h and the concentration of DMSO did not exceed 0.1% (v/v). Fluorescence was measured using a FLUOstar OPTIMA plate reader (BMG LABTECH), with OPTIMA version 2.20R2 and MARS Data Analysis Software version 3.01 R2 (BMG LABTECH). IC$_{50}$ values were determined by fitting a sigmoidal curve to the data (SigmaPlot versions 11.0 and 14.0, Systat Software): $y = a/(1 + (x/x_0)^b)$, where $y$ is '% parasite proliferation', $x$ is the concentration of the test compound, $x_0$ is the IC$_{50}$ value, and $a$ and $b$ are fitted constants.

**T. gondii culture**
*T. gondii* parasites were cultured in human foreskin fibroblasts in a humidified 37 °C incubator containing 5% CO$_2$. Infected host cells were cultured in Dulbecco's modified Eagle's medium (DMEM) containing 2 g/L sodium bicarbonate and supplemented with 1% v/v foetal calf serum, 50 U/ml penicillin, 50 μg/ml streptomycin, 10 μg/ml gentamicin, 0.25 μg/ml amphotericin B, and 0.2 mM L-glutamine.

**Generating T. gondii parasites with a G419S mutation in TgATP4**
To generate a strain of *T. gondii* parasites wherein the *tgatp4* gene was mutated to express a TgATP4 protein containing the G419S mutation, we used a CRISPR-Cas9 genome editing approach. First, we designed a single guide RNA (gRNA) targeting the *tgatp4* genomic locus near the region encoding amino acid residue 419. We introduced this gRNA-encoding sequence into the pSAG1::Cas9-U6::sgUPRT plasmid (Addgene plasmid 54467) using Q5 mutagenesis (New England Biolabs)[68]. For the Q5 mutagenesis, we used primers 12 and 13 (Supplementary Table 7). We also generated a donor DNA consisting of the annealed primers 14 and 15 (Supplementary Table 7), which encodes for a region homologous to the *tgatp4* locus that incorporates amino acid reside 419. To anneal these primers, we combined them at a 50 μM final concentration, heated them to 94 °C in a heat block for 10 min, then allowed the heat block to cool to room temperature across several hours. We mixed the sgRNA-expressing plasmid, which also encodes Cas9-GFP, together with the donor DNA, and transfected the mixture into TgATP4-HA strain parasites, which were generated previously by fusing the TgATP4 protein to an HA-epitope tag in TATi/Δku80 parasites[18]. We selected and cloned GFP-expressing parasites two days after transfection using fluorescence activated cell sorting[69]. To identify parasite clones encoding the G419S mutation in the TgATP4 protein, we PCR amplified the *tgatp4* genomic locus from clonal parasite lines using primers 16 and 17 (Supplementary Table 7), and subjected PCR products to Sanger sequencing. We identified multiple clones containing the G419S mutation (Supplementary Fig. 4) and, following verification that the expression levels and localisations of the TgATP4$^{G419S}$-HA protein did not differ between the various clones (Supplementary Fig. 5), used clone B9 for subsequent experiments.

**Determination of the expression and localisation of TgATP4$^{G419S}$**
To compare the expression levels of the TgATP4$^{G419S}$-HA protein to that of TgATP4$^{WT}$-HA, we extracted proteins from the various parasite lines in LDS sample buffer (Thermo Fisher Scientific), loaded these at 2.5 × 10$^6$ parasite equivalents per lane, and separated them by SDS-PAGE using Bis/Tris NuPAGE gels (12% acrylamide, Thermo Fisher Scientific) according to the manufacturer's instructions. Samples were then transferred to nitrocellulose membranes using the Mini Blot module (Thermo Fisher Scientific), and subjected to western blotting. We probed blots with rat anti-HA (clone 3F10, Sigma catalogue number 11867423001, diluted 1:500) and anti-TgTom40[70] (diluted 1:2000) primary antibodies, and horseradish peroxidase-conjugated goat anti-rat (Abcam, ab97057, diluted 1:5000) and goat anti-rabbit (Abcam,

ab97051, diluted 1:10000) secondary antibodies. Blots were developed using a homemade chemiluminescence solution (0.04% w/v luminol, 0.007% w/v coumaric acid, 0.01% v/v $H_2O_2$, 100 mM Tris pH 9.4) and imaged using X-ray film. An uncropped, unprocessed scan of the western blot is shown in the Source Data file.

To compare the localisation of the TgATP4$^{G419S}$-HA protein to that of the TgATP4$^{WT}$-HA protein, we undertook immunofluorescence assays as described previously[71]. We probed samples with rat anti-HA (clone 3F10, Sigma catalogue number 11867423001, diluted 1:500) and mouse anti-*Tg*P30 (clone TP3, Abcam, ab8313, diluted 1:500) primary antibodies, and donkey anti-rat AlexaFluor 488 (Thermo Fisher Scientific, A-21208, diluted 1:500) and goat anti-rabbit AlexaFluor 546 (Thermo Fisher Scientific, A-11035, diluted 1:500) secondary antibodies. Samples were imaged using an Olympus IX71 microscope featuring a DeltaVision Elite set-up, with a 100X UPlanSApo objective lens (NA 1.40) and a monochrome CoolSNAP HQ2 camera. Images were deconvolved using SoftWoRx Suite version 2.0 software, pseudo-coloured, and adjusted linearly for contrast and brightness.

## Measurements of [Na$^+$]$_{cyt}$

Saponin-isolated mature trophozoite-stage *P. falciparum* parasites, or extracellular *T. gondii* tachyzoites, were loaded with the Na$^+$-sensitive fluorescent dye SBFI (6 µM; in the presence of 0.01% w/v Pluronic F-127) for 20 min (*P. falciparum*) or ~30 min (*T. gondii*)[11,18]. Measurements were performed with parasites suspended at 37 °C in pH 7.1 'Physiological Saline Solution' (125 mM NaCl, 5 mM KCl, 1 mM MgCl$_2$, 20 mM glucose and 25 mM HEPES)[11,18,19]. The ratio of the fluorescence intensity recorded at 340 nm and 380 nm (with an emission wavelength of 515 nm) was converted to [Na$^+$]$_{cyt}$ using a calibration procedure that entailed suspending parasites in solutions of varying [Na$^+$] containing ionophores[11]. Fluorescence measurements and calibrations were performed at 37 °C, either with individual 1 mL suspensions using a PerkinElmer LS 50B fluorescence spectrometer (for resting [Na$^+$]$_{cyt}$ measurements for which the data are shown in Supplementary Fig. 7) or in 96-well plates using a Tecan Infinite M1000 PRO plate reader with i-control (version 1.12) software (all other [Na$^+$]$_{cyt}$ measurements).

## Measurements of membrane ATPase activity

Membranes from isolated *P. falciparum* or extracellular *T. gondii* parasites were prepared by lysing parasites in ice-cold deionized water containing a 1:500 dilution of Protease Inhibitor Cocktail Set III (Merck Millipore), then washing the membrane preparation three times in ice-cold deionized water (with protease inhibitors present for the first two washes)[15]. Protein concentrations in the membrane preparations were measured using a Bradford assay[72]. The PiColorLock Gold Phosphate Detection System (Innova Biosciences) was used to measure the production of Pi from hydrolysis of ATP[15]. Membrane preparations were diluted in either a high Na$^+$ solution (yielding final conditions in the reactions of 150 mM NaCl, 20 mM KCl, 2 mM MgCl$_2$, 50 mM Tris, pH 7.2) or a Na$^+$-free solution (yielding final conditions of 150 mM choline chloride, 20 mM KCl, 2 mM MgCl$_2$, 50 mM Tris, pH 7.2) to achieve a total protein concentration of 50 µg/mL. Cipargamin, (+)-SJ733, or DMSO (solvent control) were added to give the concentrations stated in the relevant Figure legends. The reactions were performed at 37 °C, and were initiated by the addition of 1 mM ATP (Na$_2$ATP·3H$_2$O; MP Biomedicals) and terminated 10 min later by adding 100 µL of reaction mixture (in duplicate) to 25 µL "Gold mix" in a 96-well plate. "Stabiliser" was added 3 min later, and the plates incubated at room temperature in the dark for 60 min before measuring absorbance at 635 nm. Background values (averaged from wells containing all components, but to which ATP was not added until after the membrane was exposed to Gold mix) were subtracted from the data. For each treatment, the ATPase activity associated with the (Na$^+$-dependent) PfATP4 and TgATP4 proteins was calculated by subtracting data obtained in the low Na$^+$ condition

(containing only the 2 mM Na$^+$ introduced on addition of 1 mM Na$_2$ATP) from that obtained in the high Na$^+$ condition.

## Comparisons of the growth rate of asexual *P. falciparum* parasites

The parasitaemias of synchronous cultures containing trophozoite-stage parasites were determined by flow cytometry. Briefly, *P. falciparum*-infected erythrocytes (1 mL) were centrifuged (2000 × *g*, 30 s), and the cells were resuspended in 1 mL of pH 7.4 Physiological Saline Solution containing 20 µg/mL Hoechst 33258. The cells were incubated for 15 min at 37 °C, and were then centrifuged and washed twice in pH 7.4 Physiological Saline Solution (2000 × *g*, 30 s) before being resuspended in 1 mL of pH 7.4 Physiological Saline Solution. The cell suspension (50 µL) was then diluted into 250 µL pH 7.4 Physiological Saline Solution to a density of ~10$^6$–10$^7$ cells/mL in 1.2 mL Costar polypropylene cluster tubes (Corning). Cells (100,000) were sampled from each tube at a low sampling speed with the following settings: forward scatter = 308 V (log scale), side scatter = 308 V (log scale), Alexa Fluor 488 = 559 V (log scale) and Pacific Blue = 478 V (log scale). The instrument used was a BD LSR II, with data collected using FlowJo version 10 (FlowJo LLC.) and BD FACSDiva version 9.0. The gating strategy is illustrated in Supplementary Fig. 10.

The volume of saponin-isolated trophozoites was measured using a Beckman Coulter Multisizer 4 (with Multisizer 4 software version 4.01; Beckman) fitted with a 100 µm 'aperture tube'[12]. The parasites were washed and resuspended (at 37 °C) in pH 7.1 Physiological Saline Solution. The electrolyte solution within the aperture tube was also pH 7.1 Physiological Saline Solution. For each measurement of cell volume, approximately 20,000 pulses (each corresponding to the passage of a single cell through the aperture) were recorded. The mean volume of the parasites within each sample was determined by fitting a Gaussian distribution curve to the population data within a 6–60 fL window in GraphPad Prism version 8.

## Gametocyte culture and mosquito infections

Using the gene-edited G358S mutants generated in the gametocyte competent NF54 line (NF54$^{G358S-1}$, NF54$^{G358S-2}$ and NF54$^{G358S-3}$), the binding-site mutant control NF54$^{CTL}$, and NF54 parental controls (NF54$^{WT}$), we prepared gametocyte cultures as described earlier[73]. Gametocyte cultures were set up in six well plates. Five mL of an asexual stage culture at 5% parasitaemia was centrifuged at 500 × *g* for 5 min at room temperature. The supernatant medium was discarded and the cells were resuspended in 30 mL of complete culture medium[73]. Packed uninfected erythrocytes (1.2 mL) were then added and the cell suspension was mixed, before 5 mL volumes were dispensed into each well of the six well plate. The culture medium was changed daily for 15-18 days, by carefully aspirating ~70–80% of the supernatant medium to avoid removing cells, and 5 mL of fresh complete culture medium was added to each well. Giemsa-stained blood smears were made every alternate day to confirm that the parasites remained viable. On day 15 to 18, gametocyte cultures were transferred to pre-warmed tubes and centrifuged at 500 × *g* for 5 min. The cells were diluted in a pre-warmed 50:50 mixture of uninfected erythrocytes and normal human serum to achieve a gametocytemia of 0.2% and the resulting 'feeding mixture' was placed into a pre-warmed glass feeder. Uninfected *Anopheles stephensi* mosquitoes were allowed to feed on the culture for 30 min. Unfed mosquitoes were discarded and the mosquito cage was placed in a humidified 26 °C incubator, with 10% sugar-soaked cotton pads placed on top of the mosquito cage.

On Day 12 post blood meal, mosquito midguts were dissected, stained with 0.2% mercurochrome, and oocysts were counted using a 4× objective. On Day 16 post blood meal, salivary glands were dissected from 20 mosquitoes (for each parasite line), pooled and homogenised, and sporozoites were counted on a haemocytometer to

determine the average number of sporozoites per mosquito for each line.

## Molecular docking

Molecular docking utilised two structures generated with AlphaFold2[45] through the ColabFold Google colabatory notebook[44]. The first was contributed by the Ersilia Open Source Initiative (https://github.com/ersilia-os/osm-pfatp4-structure) and utilised HHblits[74] in the generation of the multiple sequence alignment. Our own Colab-Fold structure used the default MMseqs2 alignment[75] produced for PfATP4. The PfATP4 sequence used was downloaded from the UniProt server (Accession ID: Q9U445; https://www.uniprot.org/uniprotkb/Q9U445/entry). The first 115 residues were removed, as they were in the Open Source protocol, as these were predicted to be disordered, leaving 1149 modelled residues. A Google pro plan was required to generate an AlphaFold2 model of that length. No templates were used and energy minimisation was performed after model generation using the Molecular Dynamics software, GROMACS 2021[76]. The highest ranking protein structure generated by ColabFold (version 1.3.0) (predicted local distance difference (pLDDT) of 84.75, compared to a pLDDT of 88.38 for the Open Source structure), was solvated with TIP3 water and energy minimised until the maximum force was below $240\,kcal\,mol^{-1}\,nm^{-1}$ using the AMBER99sb forcefield[77]. Backbone atoms were restrained. The G358S mutation was generated by using the pdb reader tool through Charmm-GUI[78].

Docking was performed using AutoDock Vina version 1.1.2[43]. Pdbqt files were generated using autodock tools. The ligand molecules cipargamin and (+)-SJ733 were downloaded from PubChem (CID 44469321 (https://pubchem.ncbi.nlm.nih.gov/compound/44469321) and 89508529 (https://pubchem.ncbi.nlm.nih.gov/compound/89508529), respectively). To thoroughly search the full surface of the protein, the search space was divided into three with regions of overlap equal to the length of the ligand. Three replicate searches were run with increasing amounts of exhaustiveness until increased searching no longer changed the top nine poses across all poses found (an accumulated exhaustiveness of 2040). For the constrained search of the potential binding cavity, a box size of 35 × 30 × 27 Å was employed as determined with the Autodock tools graphical user interface (GUI). Three replicates with an exhaustiveness of 1024 were run to produce consistent results among replicates. Results were visualised in VMD (version 1.9.3)[79] with the aid of custom tcl scripts.

## Statistics and reproducibility

The statistical tests used in this study were paired *t* tests, unpaired *t* tests and Mann-Whitney tests (two-tailed in all cases; performed with GraphPad Prism versions 8 and 9). For each data set, the test that was used is indicated in the legend of the relevant figure or table. Sample size was not predetermined using statistical methods. At least three independent experiments were performed for all parasite proliferation assays and biochemical assays. The western blot and immunofluorescence assay shown in Supplementary Fig. 5 were only performed once; however, the experiments yielded the same result for multiple clones and the expression and correct localisation of TgATP4 was also evidenced from biochemical assays that were performed at least three times. The figure legends and tables state the exact number of biological replicates that were performed. All the experimental findings were found to be highly reproducible in biological replicates, including when these were performed by different researchers (as was the case for all key findings). Data were only excluded in rare instances in which a technical error took place during the execution of an experiment. In biochemical assays and parasite proliferation assays involving different compounds and/or parasite lines, there was randomisation with respect to where these were positioned on the assay plates. Other experiments were not randomised. The Investigators were not blinded to allocation during experiments and outcome assessment.

## Reporting summary

Further information on research design is available in the Nature Research Reporting Summary linked to this article.

## Data availability

The whole-genome sequencing data for this study have been deposited in the European Nucleotide Archive (ENA) at EMBL-EBI under accession numbers PRJEB53576 (HCR and parental lines; https://www.ebi.ac.uk/ena/browser/view/PRJEB53576) and PRJEB55457 (NF54-based lines; https://www.ebi.ac.uk/ena/browser/view/PRJEB55457). The structural models of PfATP4 generated in this study are available in GitHub (https://github.com/CorryLab/PfATP4_Colabfold-models). The 3D7 reference genomes used in the analyses are available in the PlasmoDB database. These were PlasmoDB-29_Pfalciparum3D7 for the HCR and parental lines (https://plasmodb.org/plasmo/app/downloads/release-29/Pfalciparum3D7/) and PlasmoDB-48_Pfalciparum3D7 for the NF54-based lines (https://plasmodb.org/plasmo/app/downloads/release-48/Pfalciparum3D7/). All other data supporting the findings of this study are available within the paper and Supplementary Information. Source data are provided with this paper.

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

## Acknowledgements

The authors are grateful to the Canberra Branch of the Australian Red Cross Lifeblood for the provision of blood, to the Medicines for Malaria Venture for the provision of several of the compounds used in this study, and to Assoc. Prof. Erkang Fan and Prof. Akhil Vaidya for the provision of PA21A050. We also thank Prof. Elizabeth Winzeler for the Dd2-PfATP4$^{T418N,P990R}$ and Dd2 parental lines, and Prof. R. Kip Guy for (+)-SJ733 and for the W2-PfATP4$^{P966S}$, W2-PfATP4$^{P966T}$ and W2 parental lines. We are grateful to Xinxin Zhang, Sasha Lee and Kwong Sum Shea for initial exploratory Na$^+$ assays and ATPase assays with the HCR and Dd2-PfATP4$^{T418N,P990R}$ lines, to Jennifer Thompson and Prof. Alan Cowman for assistance with whole-genome sequencing, and to Michael Devoy (Imaging & Cytometry Facility, The John Curtin School of Medical Research, Australian National University) for assistance with flow cytometry. The authors would also like to thank Dr Miquel Duran-Frigola and the Ersilia Open Source Initiative for access to their PfATP4 AlphaFold2 model, Dr Thierry Diagana and Dr Caroline Boulton for helpful comments on the manuscript, the Bloomberg Family Foundation for their generous support of the Insectary and Parasitology core facilities at the Johns Hopkins Malaria Institute, Chris Kizito and Godfree Mlambo for expert mosquito rearing and core facility management, and Sachie Kanatani for help with the photographs of the mosquito midguts. This work was supported by: an Australian National Health and Medical Research Council Project Grant (GNT1159648) to A.M.L. and K.K.; a contract from Novartis to D.A.F., P.S. and A.K.T.; a Wellcome grant [Grant number 206194/Z/17/Z] to M.C.S.L.; and a National Institutes of Health grant (R01 AI132359) to P.S.

## Author contributions

Performed and analysed experiments—D.Q., J.V.P., J.E.O.R., V.T., D.L., Y.X., Y.T.V.A., J.Y.H.A., G.X., A.K.T., N.F.G., K.J.F., B.H.S., H.H., A.S.M.D., M.C.R., G.G.v.D., A.M.L.; analysed whole-genome sequencing data—J.S.P., T.Y.; molecular docking—J.D.T., B.C.; contributed resources or methodology—K. Kümporsnin, M.C.S.L., J.M.M.; supervision—A.M.L., K. Kirk, G.G.v.D., D.A.F., P.S., B.C., A.T.P.; funding acquisition—A.M.L., K. Kirk, D.A.F., P.S., A.K.T., J.S., E.K.S.; design, coordination and planning—A.M.L., K. Kirk., D.A.F., P.S., J.S., E.K.S.; contributed to writing—K. Kirk, G.G.v.D., D.A.F., V.T., P.S., B.C., A.T.P., J.S., E.K.S., B.H.S., J.S.P., J.D.T., J.V.P., D.Q.; wrote the original draft—A.M.L.

## Competing interests

E.K.S. and J.S. are employees and shareholders of Novartis, which funded aspects of the work. The other authors have no competing interests to declare.

## Additional information

[1]Research School of Biology, Australian National University, Canberra, ACT 2600, Australia. [2]Department of Microbiology & Immunology, Columbia University Irving Medical Center, New York, NY 10032, USA. [3]Bioinformatic Division, The Walter & Eliza Hall Institute of Medical Research, Parkville, VIC 3052, Australia. [4]Department of Molecular Microbiology & Immunology and Johns Hopkins Malaria Institute, Johns Hopkins School of Public Health, Baltimore, MD 21205, USA. [5]Wellcome Sanger Institute, Wellcome Genome Campus, Hinxton CB10 1SA, UK. [6]Novartis Pharma AG, Novartis Campus, Basel 4056, Switzerland. [7]Novartis Institute for Tropical Diseases, Emeryville, CA 94608, USA. [8]Department of Medical Biology, The University of Melbourne, Parkville, VIC 3052, Australia. [9]Center for Malaria Therapeutics and Antimicrobial Resistance, Division of Infectious Diseases, Department of Medicine, Columbia University Irving Medical Center, New York, NY 10032, USA. [10]These authors contributed equally: Deyun Qiu, Jinxin V. Pei, James E. O. Rosling. ✉e-mail: adele.lehane@anu.edu.au

