## [Peer review file · Nature Communications]

REVIEWERS' COMMENTS

Reviewer #1 (Remarks to the Author):

The paper describes selection of resistance to cipargamin, a new antimalarial currently undergoing development at the clinical trial stage. Malaria parasite resistance to antimalarials is an ongoing problem in controlling disease and a pipeline of new drugs needs to be maintained to backfill the loss of previous drugs that have succumbed to resistance. Prior knowledge of the options parasites have to getting around new drugs is crucial to successful deployment aimed at maximising the efficacy window such that the pipeline does not run dry. This study is therefore of considerable importance given that cipargamin is hoped to replace the current frontline antimalarial artemisinin, resistance to which is spreading globally.

The paper describes various methods of resistance selection, using various genetic backgrounds, that repeatedly identify a mutation (G385S) in a gene known as PfATP4. Previous work shows PfATP4 is the primary target of compounds like cipargamin and this study shows the G385S mutation confers 100 fold to 1000 fold resistance to cipargamin depending on background. Proof that the G385S mutation in PfATP4 is sufficient to confer cipargamin resistance is elegantly demonstrated by knocking the mutation into clean/sensitive backgrounds using CRISPR/Cas9 genome editing. Essentially, the point mutation in PfATP4 will render the drug useless and G385S is shown to be a hotspot for resistance to cipargamin and related inhibitors.

Trials of the resistant mutants with other drugs reveal that the G385S mutation in PfATP4 confers some cross resistance to cipargamin-like inhibitors but not unrelated compounds with different targets, suggesting that the mechanism of resistance is indeed target based rather than alternative mechanisms that might sequester, neutralise or expel the inhibitor. Direct interaction of cipargamin with PfATP4 is further implicated by docking of the inhibitor with AlphaFold models of the protein in which the pocket is less amenable to inhibitor binding when G385 is changed to 385S.

The paper then examines the evolutionary fitness consequences of the G385S mutation in PfATP4 on malaria parasites. Elevated cytosolic Na⁺ levels are inferred in resistant mutants via reduced ATP consumption in membrane fractions, which is accepted as a proxy for the action of ATP4 in expelling Na⁺. A similar outcome is found in mutants of the orthologue of the related parasite *Toxoplasma gondii*. Asexual malaria parasite growth in PfATP4 mutants is unequivocally shown to be equivalent to wild type, identifying no fitness cost during in vitro replication. The PfAT4 mutants are also shown to infect laboratory mosquitoes efficiently, which likely means that the resistance mutants will transmit in the wild.

All in all, the work is thoroughly and competently executed. The figures and tables are all clear and well explained. The conclusions are justified, and the findings warrant some caution being exercised in the deployment of cipargamin as an antimalarial since resistance seems facile for the parasite to acquire and has no obvious deleterious consequences to parasite fitness.

I recommend some minor changes –

Line 36 (and again in line 581). The findings DO NOT suggest that “a drug combination strategy would be a suitable risk mitigation strategy”. The findings very strongly suggest that a risk mitigation strategy should be put in place, but they don’t address whether drug combination will mitigate risk. Logically it should, but none of the data herein support that approach. Reword to be more circumspect

Line 447. Please explain why extended in vitro culture might impact mosquito infection intensity in the CRISPR/Cas synonymous control lines. Do the authors think that the control line has acquired an arbitrary transmission defect during asexual passage? Anything in the genome sequence to suggest that?

Line 540. There is no such thing as “significant homology”. Homology is an absolute state, like pregnancy or virginity (see Lewin, R., When does homology mean something else? *Science*, 1987. 237: 1570. or Reeck et al., "Homology" in proteins and nucleic acids: a terminology muddle and a way out of it. *Cell*, 1987. 50: 667.

Reviewer #2 (Remarks to the Author):

The MS "A G358S mutation in the Plasmodium falciparum Na⁺ pump PfATP4 confers clinically relevant levels of resistance to cipargamin and (+)-SJ733" by D Qiu, JV Pei, K Kirk, and AM Lehane et al presents details of a notably thorough and carefully conducted examination relating to the pinpointing of resistance recorded for two lead antimalarial compounds, the spiroindolone KAE609 (cipargamin) and the structurally strikingly disparate substituted tetrahydroisoquinol-1-one (+)-SJ733. The authors note and partially cite relevant work indicating that spiroindolones interfere with Na⁺ transport into the parasite cytosol by PfATP4, and that mutations in the transporter protein confer resistance to the effects of this drug class. For both cipargamin and (+)-SJ733, it is noted in a review (not cited in this MS) that such mutations confer cross resistance in vitro in response to drug pressure (Luth, Winzeler et al. 2018 Using in vitro evolution and whole genome analysis to discover next generation targets for antimalarial drug discovery. *ACS Infect. Dis.* 4, 301–314; doi: 10.1021/acsinfecdis.7b00276).

Of immediate relevance to the current MS, it has been previously reported that the G358S mutation confers clinically significant resistance against cipargamin in clinical trials (Schmitt, Gandhi et al., ref. 39). For (+)-SJ733, although the relevant work is cited later in the MS, mutations in the same region of the transporter protein also as introduced via CRISPR/Cas9 gene editing – principally noted as L350H, although G358S is also noted - also elicits resistant in PfATP4 (Jimenez-Diaz, Guy et. al., ref. 23; Crawford, Quan, de Risi et al, ref. 48). In other words, with the exception of the significance of the specific mutation G358S with respect to resistance to (+)-SJ733, the areas including the techniques and covered by the MS have been thoroughly and carefully discussed previously.

The authors' methods include raising the drug-resistant Pf parasites including parasite bearing the G358S PfATP4 mutation according to previously published methods, or as provided by lead investigators associated with the foregoing references, whole genome sequencing, and parasite proliferation assays. Measurement of intracytosol Na⁺ concentration, and rate comparisons of wild-type and genetically modified parasites are carried out. The structural effects conferred by the G358S mutation were explored by using DeepMind's AlphaFold and two structures to create a protein structure containing the mutation wherein it was noted that 'steric clashes' involving each of cipargamin and (+)-SJ733 with the S358 side chain inhibited high affinity binding. It is unsure which structures were used, as a search for ATP4 (AlphaFold Protein Structure Database (ebi.ac.uk)) turns up structures of ATP4-ase for manifold organisms, but not for Pf. The website cited in ref. 44 directs to a structure generated by AlphaFold itself, and as used in the current MS.

It is emphasized that the MS is very well written and contains singularly high-quality work that is carefully carried out and which provides results that are compatible with the conclusions. Clearly this is worthy of publication as a full paper in a high impact factor journal. However, given especially the quality of the earlier work, in particular, that by Schmitt, Gandhi et al., ref. 39, in originally pinpointing the G358S mutation and how this is clearly associated with deep-seated clinical failures (the recrudescence data in particular is impressive), and for (+)-SJ733, the CRISPR/Cas9 gene editing leading to a mutant eliciting resistance in PfATP4 (Jimenez-Diaz, Guy et. al., ref. 23; Crawford, Quan, de Risi et al, ref. 48) must take priority. In particular the structural elucidation of the mutant PfATP4 protein and the location of the binding site with (+)-SJ733 is impressive, given the structural data available at the time of publication of ref. 23 (2014). Overall, unfortunately, the current MS lacks the originality and impact to justify publication in Nature Communications.

Our responses to Reviewer 1's comments

Reviewer 1 Point 1: Line 36 (and again in line 581). The findings DO NOT suggest that “a drug combination strategy would be a suitable risk mitigation strategy”. The findings very strongly suggest that a risk mitigation strategy should be put in place, but they don't address whether drug combination will mitigate risk. Logically it should, but none of the data herein support that approach. Reword to be more circumspect

Our response to Reviewer 1 Point 1:

The data that we were drawing on when making this suggestion was our finding that PfATP4-G358S parasites remain sensitive to antimalarials that do not target PfATP4. Nonetheless, we take the Reviewer's point that this should be reworded to be more circumspect.

We have changed the relevant sentence in the Abstract (lines 36-39; changes underlined) from:

“Our findings suggest that PfATP4 inhibitors in clinical development should be tested against PfATP4^{G358S} parasites, and that a drug combination approach would be a suitable risk mitigation strategy.”

to:

“Our findings suggest that PfATP4 inhibitors in clinical development should be tested against PfATP4^{G358S} parasites, and that their combination with unrelated antimalarials may mitigate against resistance development.”

We have changed the relevant sentence in the Discussion (lines 592-595 in revised manuscript; changes underlined) from:

“Importantly, a drug combination approach would be a suitable risk mitigation strategy for PfATP4 inhibitors, since parasites with the G358S mutation in PfATP4 remained fully susceptible to other antimalarials with unrelated modes of action.”

to:

“Importantly, a drug combination approach may serve as a suitable risk mitigation strategy for PfATP4 inhibitors, since parasites with the G358S mutation in PfATP4 remained fully susceptible to other antimalarials with unrelated modes of action.”

Reviewer 1 Point 2: Line 447. Please explain why extended *in vitro* culture might impact mosquito infection intensity in the CRISPR/Cas synonymous control lines. Do the authors think that the control line has acquired an arbitrary transmission defect during asexual passage? Anything in the genome sequence to suggest that?

Our response to Reviewer 1 Point 2:

Prolonged *in vitro* culture of asexual blood stages can result in the acquisition of mutations or gene copy number variants that negatively impact traits that are essential for parasite survival *in vivo* but dispensable *in vitro*, e.g. transmissibility to mosquitoes or cytoadhesion (e.g. Kemp DJ, Thompson J, Barnes DA, Triglia T, Karamalis F, Petersen C, Brown GV, Day KP. A chromosome 9 deletion in

Plasmodium falciparum results in loss of cytoadherence. Mem Inst Oswaldo Cruz. 1992;87(Suppl 3):85–89. doi: 10.1590/S0074-02761992000700011; Alano P, Roca L, Smith D, Read D, Carter R, Day K. *Plasmodium falciparum*: parasites defective in early stages of gametocytogenesis. Exp Parasitol. 1995;81:227–235; Pologe LG, Ravetch JV. Large deletions result from breakage and healing of *P. falciparum* chromosomes. Cell. 1988 Dec 2;55(5):869–74. doi: 10.1016/0092-8674(88)90142-0. PMID: 3056622.) Therefore, we analyzed the edited parasite genomes by whole-genome sequencing (WGS) and compared those data to that of the NF54^{WT} parental strain to search for genetic markers associated with putative altered parasite transmissibility, particularly in the edited control line NF54^{CTL}. These data were presented in Data S2 (includes Tables S2-S4) as part of the Supplementary Information. We have now moved the paragraph that discusses the WGS findings from Data S2 (pages 6-7) to the relevant section in the Results. Revised paragraphs (with changed and new text underlined) now read as follows:

Lines 443-446 of revised manuscript: “While the infection intensities for the NF54^{G358S} lines and the NF54^{CTL} line were somewhat lower than those for the NF54^{WT} line, parasites from each of the lines formed healthy-appearing oocysts (Supplementary Fig. 8), indicating that the G358S mutation in PfATP4 did not affect the establishment of infection in mosquitoes.”

Lines 455-468 of revised manuscript: Edited parasite genomes were analyzed by whole-genome sequencing and compared to that of the NF54^{WT} parental line to search for genetic changes that might have spontaneously arisen during prolonged *in vitro* culture and associate with altered transmissibility to mosquitoes (Supplementary Tables 2-4). We confirmed the presence of the PfATP4 G358S mutation in 100% of the parasite population in the two gene edited mutant lines, NF54^{G358S-1} and NF54^{G358S-2}, along with the presence of 17 silent binding-site mutations at the *pfatp4* guide RNA cleavage site that were introduced as part of the CRISPR-Cas9 editing strategy. These silent mutations were also present in the edited control line NF54^{CTL} that expresses wild-type PfATP4 (Supplementary Tables 3, 4). We also identified two additional high confidence SNPs coding for non-synonymous mutations that were absent in the NF54^{WT} parental strain: (1) a SNP in PF3D7_0304000, coding for a V78A change in inner membrane complex protein 1a, present in both NF54^{CTL} and the PfATP4 mutant line NF54^{G358S-2}; and (2) a SNP in PF3D7_1251500, coding for a N521Y change in the ATP-dependent RNA helicase DRS1, found only in NF54^{G358S-1} (Supplementary Tables 3, 4). No common SNP was found in the PfATP4-edited lines that could explain the slightly lowered infection intensities and sporozoite loads compared with NF54^{WT} parasites. We also observed no CNVs in any of the edited parasite lines.”

Reviewer 1 Point 3: Line 540. There is no such thing as “significant homology”. Homology is an absolute state, like pregnancy or virginity (see Lewin, R., When does homology mean something else? Science, 1987. 237: 1570. or Reeck et al., "Homology" in proteins and nucleic acids: a terminology muddle and a way out of it. Cell, 1987. 50: 667.

Our response to Reviewer 1 Point 3:

We have removed the word ‘significant’ before ‘homology’.

Our responses to Reviewer 2’s comments

Reviewer 2 Point 1: The authors note and partially cite relevant work indicating that spiroindolones interfere with Na⁺ transport into the parasite cytosol by PfATP4, and that mutations in the transporter protein confer resistance to the effects of this drug class. For both cipargamin and (+)-

SJ733, it is noted in a review (not cited in this MS) that such mutations confer cross resistance in vitro in response to drug pressure (Luth, Winzeler et al. 2018 Using in vitro evolution and whole genome analysis to discover next generation targets for antimalarial drug discovery. ACS Infect. Dis. 4, 301–314; doi: 10.1021/acscinfecdis.7b00276).

Our response to Reviewer 2 Point 1:

We agree that it is worth noting in the manuscript that cross-resistance to different PfATP4 inhibitors has been seen for various PfATP4 mutant parasites previously. We have added an additional sentence to the Introduction. The relevant section (lines 111-112 in revised manuscript; new text underlined) now reads as follows:

“More than 40 different resistance-associated SNPs have been reported in *pfatp4*, with most individual parasite lines having a single mutation in PfATP4, a small number having two mutations, and one reported to have three mutations (9,20,21,23-25,35). Cross-resistance to structurally unrelated PfATP4 inhibitors has been demonstrated for multiple distinct PfATP4 mutant parasites (20,21,23,24,36).”

The references supporting the new sentence include four primary research articles in which resistance to multiple PfATP4 inhibitors has been shown for one or more PfATP4 mutant parasite(s), as well as the Luth et al. review (ref. 36 in the revised manuscript).

Reviewer 2 Point 2: Of immediate relevance to the current MS, it has been previously reported that the G358S mutation confers clinically significant resistance against cipargamin in clinical trials (Schmitt, Gandhi et al., ref. 39). For (+)-SJ733, although the relevant work is cited later in the MS, mutations in the same region of the transporter protein also as introduced via CRISPR/Cas9 gene editing – principally noted as L350H, although G358S is also noted - also elicits resistant in PfATP4 (Jimenez-Diaz, Guy et. al., ref. 23; Crawford, Quan, de Risi et al, ref. 48). In other words, with the exception of the significance of the specific mutation G358S with respect to resistance to (+)-SJ733, the areas including the techniques and covered by the MS have been thoroughly and carefully discussed previously.

Our response to Reviewer 2 Point 2:

The only published reports relating to the G358S mutation in PfATP4 thus far are:

1. Its appearance after an *in vitro* selection with (+)-SJ733 (Jimenez-Diaz et al. 2014). The PfATP4-G358S line generated in this study was reported to have only a low level of resistance to (+)-SJ733. No further experiments with the line were reported. In our study we show that this line actually has a very high level of resistance to both cipargamin and (+)-SJ733.
2. Its presence in recrudescence parasites in a recent clinical trial for cipargamin (Schmitt et al. 2021). The recrudescence PfATP4-G358S parasites were not characterised in any way in this study.

To date, there have been no published reports of the effect of the G358S mutation in PfATP4 on any aspect of parasite physiology or fitness, no testing of its effects on parasite susceptibility to PfATP4 inhibitors (except for the conflicting data pertaining to (+)-SJ733 in Jimenez-Diaz et al. 2014, as above) or unrelated antimalarials, and no assessment of its effects on PfATP4 function. We also note that gene-editing to introduce the G358S mutation in PfATP4 has not been performed to date in any published study.

As such, our study is crucial for understanding the risks associated with this clinically critical mutation, its effects on the parasite, and the mechanistic basis for its effects.

Reviewer 2 Point 3: It is unsure which structures were used, as a search for ATP4 (AlphaFold Protein Structure Database (ebi.ac.uk)) turns up structures of ATP4-ase for manifold organisms, but not for Pf. The website cited in ref. 44 directs to a structure generated by AlphaFold itself, and as used in the current MS.

Our response to Reviewer 2 Point 3:

The two structural models presented in our study were generated with ColabFold (which utilises AlphaFold2). We have clarified this in the manuscript. The PfATP4 structure generated as part of the AlphaFold2 Protein Database (Varadi et al. 2022 Nucleic Acids Res) was also examined, however it was of lower quality (with a lower pLDDT: 82.64) than our ColabFold models (pLDDTs: 84.75 and 88.38). We have therefore not incorporated it into the study.

We have changed the relevant section in the Results (lines 344-346 in revised manuscript; changes underlined) from:

“As no atomic resolution structures are available for PfATP4 we utilised two structures generated by AlphaFold2 (43), one created by us and one contributed to Open Source Malaria by the Ersilia Open Source Initiative”

to:

“As no atomic resolution structures are available for PfATP4 we utilised two structures generated using ColabFold (44) (which utilises AlphaFold2 (45)), one created by us and one contributed to Open Source Malaria by the Ersilia Open Source Initiative”

We have changed the relevant sentence in the Discussion (lines 540-542 in revised manuscript; changes underlined) from:

“Molecular docking studies with two AlphaFold2 models of PfATP4 provided a possible mechanistic explanation, suggesting that the G358S mutation creates a steric clash that reduces the binding affinity of cipargamin and (+)-SJ733.”

to:

“Molecular docking studies with two ColabFold models of PfATP4 provided a possible mechanistic explanation, suggesting that the G358S mutation creates a steric clash that reduces the binding affinity of cipargamin and (+)-SJ733.”

Reviewer 2 Point 4: However, given especially the quality of the earlier work, in particular, that by Schmitt, Gandhi et al., ref. 39, in originally pinpointing the G358S mutation and how this is clearly associated with deep-seated clinical failures (the recrudescence data in particular is impressive), and for (+)-SJ733, the CRISPR/Cas9 gene editing leading to a mutant eliciting resistance in PfATP4 (Jimenez-Diaz, Guy et. al., ref. 23; Crawford, Quan, de Risi et al, ref. 48) must take priority. In particular the structural elucidation of the mutant PfATP4 protein and the location of the binding site with (+)-SJ733 is impressive, given the structural data available at the time of publication of ref. 23 (2014). Overall, unfortunately, the current MS lacks the originality and impact to justify publication in Nature Communications.

Our response to Reviewer 2 Point 4:

Please see our response to Reviewer 2 Point 2.